# Research and analysis of differential gene expression in CD34 hematopoietic stem cells in myelodysplastic syndromes

**Min-xiao Wang[1,2,3]‡, Chang-sheng Liao[2,4]‡, Xue-qin Wei[1,2], Yu-qin Xie[1,2], Peng-fei Han[4]\*, Yan-hui Yu[1,3]\***

1 Department of Hematology, Heping Hospital Affiliated to Changzhi Medical College, Changzhi Medical College, Changzhi, Shanxi, China, 2 Department of Graduate School, Graduate Student Department of Changzhi Medical College, Changzhi, Shanxi, China, 3 The Stem Cell and Tissue Engineering Research Center, Changzhi Medical College, Changzhi, Shanxi, China, 4 Department of Orthopedics, Heping Hospital Affiliated to Changzhi Medical College, Changzhi, Shanxi, China

‡ Min-xiao Wang and Chang-sheng Liao are considered as co-first authors
\* 18003551149@163.com (PFH); amilyfish@126.com (YHY)

## Abstract

### Objective

This study aims to investigate and analyze the differentially expressed genes (DEGs) in CD34 + hematopoietic stem cells (HSCs) from patients with myelodysplastic syndromes (MDS) through bioinformatics analysis, with the ultimate goal of uncovering the potential molecular mechanisms underlying pathogenesis of MDS. The findings of this study are expected to provide novel insights into clinical treatment strategies for MDS.

### Methods

Initially, we downloaded three datasets, GSE81173, GSE4619, and GSE58831, from the public Gene Expression Omnibus (GEO) database as our training sets, and selected the GSE19429 dataset as the validation set. To ensure data consistency and comparability, we standardized the training sets and removed batch effects using the ComBat algorithm, thereby integrating them into a unified gene expression dataset. Subsequently, we conducted differential expression analysis to identify genes with significant changes in expression levels across different disease states. In order to enhance prediction accuracy, we incorporated six common predictive models and trained them based on the filtered differential gene expression dataset. After comprehensive evaluation, we ultimately selected three algorithms—Lasso regression, random forest, and support vector machine (SVM)—as our core predictive models. To more precisely pinpoint genes closely related to disease characteristics, we utilized the aforementioned three machine learning methods for prediction and took the intersection of these prediction results, yielding a more robust list of genes associated with disease features. Following this, we conducted in-depth analysis of these key genes in the training set and validated the results independently using the GSE19429 dataset. Furthermore, we performed differential analysis of gene groups, co-expression analysis, and enrichment analysis to delve deeper into the mechanisms

**Data availability statement:** all relevant analysis scripts and steps have been uploaded to a GitHub repository and have obtained a permanent access link through the Figshare platform (10.6084/m9.figshare.27276612). This repository contains detailed descriptions and codes for all key steps, including data pre-processing, gene feature analysis, and result visualization.

**Funding:** This study was supported by grants from the National Natural Science Foundation of China (82300209), the Natural Science Foundation for Young Scientists of Shanxi Province (20210302124089), and the Heping Hospital Affiliated to Changzhi Medical College (Institute Level Research Fund; grant no. 2020-22).

**Competing interests:** The authors have declared that no competing interests exist.

underlying the roles of these genes in disease initiation and progression. Through these analyses, we aim to provide new insights and foundations for disease diagnosis and treatment. Figure illustrates the data preprocessing and analysis workflow of this study.

## Results

Our analysis of differentially expressed genes (DEGs) in CD34+ hematopoietic stem cells (HSCs) from patients with myelodysplastic syndromes (MDS) revealed significant differences in gene expression patterns compared to the control group (individuals without MDS). Specifically, the expression levels of two key genes, IRF4 and ELANE, were notably downregulated in CD34+ HSCs of MDS patients, indicating their downregulatory roles in the pathological process of MDS

## Conclusion

This study sheds light on the potential molecular mechanisms underlying MDS, with a particular focus on the pivotal roles of IRF4 and ELANE as key pathogenic genes. Our findings provide a novel perspective for understanding the complexity of MDS and exploring therapeutic strategies. They may also guide the development of precise and effective treatments, such as targeted interventions directed against these genes

## Introduction

Myelodysplastic Syndromes (MDS) represent a highly heterogeneous group of diseases characterized by abnormal bone marrow cellular development [1,2], ineffective hematopoiesis, cytopenia, and a notable risk of transformation into acute myeloid leukemia (AML) [3,4]. With an annual incidence rate ranging from 2.1 to 12.6 per 100,000 individuals, the prevalence significantly escalates among individuals over 70 years old, reaching 25 times that of the general population [5,6]. Notably, in the Asia-Pacific region, MDS patients account for more than half of the global caseload, and China, too, faces a persistently high incidence rate [7]. Mortality among MDS patients varies according to disease severity and treatment response, with lower risks for those with mild symptoms and favorable therapeutic outcomes, and higher risks for those with rapidly progressing disease or severe complications [8]. The intricate pathogenesis of MDS encompasses extensive involvement of genetics, epigenetics, immunology, and environmental factors, collectively contributing to the imbalance of the bone marrow microenvironment and the impairment of hematopoietic stem and progenitor cells (HSPCs) function [9,10]. Clinically, severe MDS patients frequently experience severe symptoms such as anemia, recurrent infections, and bleeding, significantly compromising their quality of life and shortening their survival period [11]. CD34, a highly specific surface marker, is widely expressed on early hematopoietic stem/progenitor cells and is vital for their identification, isolation, and investigation [12] Aberrant proliferation, differentiation defects, or apoptotic imbalance in CD34+ cells are considered pivotal in the pathogenesis of MDS, directly correlating with abnormalities in bone marrow hematopoiesis and the diversity of disease phenotypes. Given the central role of CD34+ cells in hematopoiesis, this study specifically focuses on gene expression changes in CD34+ hematopoietic stem cells from MDS patients [13,14,15].Although the etiology of MDS remains incompletely understood, the rapid development of advanced technologies such as high-throughput sequencing has enabled the identification of an increasing number of MDS-related genetic mutations. These discoveries have paved new avenues for molecular diagnosis, classification, and prognostic assessment of

MDS [16,17]. Concurrently, the emergence of immunomodulatory agents, hypomethylating agents, and targeted therapies has offered new therapeutic options and hope for MDS patients. Utilizing bioinformatics approaches, we aim to delve into biomarkers and disease-related genes intimately associated with the pathological processes of MDS. Against this backdrop, our study endeavors to contribute to precision medicine in MDS by deeply exploring gene expression changes in CD34+ cells. We anticipate that this research will not only provide novel insights into the pathogenesis of MDS but also establish a robust molecular foundation for early disease diagnosis, the formulation of precision treatment strategies, and the improvement of prognostic assessment systems. To clearly elucidate the entire research process of this study, we have provided Figure 1, which is a flowchart detailing the various key steps of the research (Fig 1).

## I. Materials and methods

### 1. Data acquisition and processing

This study primarily utilized microarray technology for gene expression analysis. We meticulously selected and obtained gene expression datasets of bone marrow CD34+ hematopoietic stem cells from patients with Myelodysplastic Syndromes (MDS) and healthy individuals from the publicly accessible Gene Expression Omnibus (GEO) database. To precisely locate the required data, we searched and extracted platform description files and series matrix files from the GEO database using keywords such as "MDS", "microarray", "human samples", and corresponding disease-specific gene expression patterns. These files contain core experimental data, including gene expression levels of the samples. Subsequently, we further processed these data files, including removing missing values and handling duplicate genes, ultimately generating a gene expression matrix file that will serve as the core input data for our subsequent analyses. We ultimately included the following four datasets for in-depth analysis: GSE81173, GSE4619, GSE19429, and GSE58831. Specifically, GSE81173 comprises 18 samples (12 disease samples and 6 control samples); GSE4619 contains 66 samples (55 disease samples and 11 control samples); GSE19429 includes 200 samples (183 disease samples and 17 control samples); and GSE58831 consists of 176 samples (159 disease samples and 17 control samples). It is noteworthy that the GSE81173 dataset originates from a Chinese laboratory, while GSE4619, GSE19429, and GSE58831 datasets all come from a UK laboratory, with the latter three datasets originating from the same research institution. Although we were unable to obtain detailed age and gender information for all samples during the data collection process, which somewhat limits the depth of our analysis, considering the primary objective of this study is to identify differentially expressed genes in bone marrow CD34+ hematopoietic stem cells in MDS and to deeply explore their crucial roles in disease progression, we have decided to include these datasets in our analysis.

### 2. Data preprocessing and identification of differentially expressed genes

To ensure the comparability of data across different samples, we first employed the normalizeBetweenArrays method to eliminate the systematic biases present in the four gene expression matrices derived from CD34+ cells of normal controls and MDS patients. This standardization step is crucial for ensuring the comparability of data across various samples. Subsequently, we utilized a Differential Expression Gene (DEG) analysis software package to compare gene expression profiles within these four standardized gene expression matrices. We chose DESeq2 as our analysis tool, which is a widely recognized and extensively validated R software package specifically designed for differential expression analysis of count data. DESeq2 models gene expression data using a negative binomial distribution and employs the

Wald test to assess whether there are significant differences in gene expression levels. When initially assessing differences in gene expression levels, we set a relatively lenient threshold for fold change, aiming to capture as many potential significantly differential genes as possible. Specifically, our selection criteria included an adjusted P-value (adjP) less than 0.05 and an absolute log2-transformed fold change ($|log2FC|$) greater than or equal to 1. These criteria were established to accurately identify gene expression differences with statistical significance, providing more valuable candidate genes for subsequent in-depth research.

## 3. Batch effect correction and differential analysis

To mitigate the potential impact of batch effects introduced by different experimental platforms, this study employed a dedicated algorithm for batch correction of the data. Specifically, we processed the normalized expression datasets from GSE81173, GSE4619, and GSE58831, which were included in the training set, using the "sva" package in R for batch effect elimination. Following batch correction, we merged the datasets under the same disease status. Subsequently, we utilized the "limma" package to perform a differential expression analysis on the combined, batch-corrected data, with the aim of identifying genes exhibiting significant differences [10]. This approach allowed us to focus on biologically meaningful variations in gene expression, rather than spurious effects arising from experimental artifacts, thereby enhancing the robustness and reliability of our findings.

## 4. Training and selection of common prediction models

For the screened differential gene expression dataset, this study aims to find the most suitable prediction model for this dataset by training and comparing various machine learning models. To this end, we have adopted the following six common machine learning models: Random Forest (RF), LASSO Regression, Support Vector Machine (SVM), Neural Network (NN), Logistic Regression (LR), and Gradient Boosting Machine (GBM).

**4.1 Overview of toolkits and functions.** To complete the model training, evaluation, interpretation, and visualization, this study uses the following R packages and their specific functions:

caret: Used for data preprocessing, model training, and performance evaluation, especially by setting cross-validation through the trainControl function.

DALEX: Provides model interpretation and evaluation functions, calculating performance metrics of different models through the model_performance function.

ggplot2: Used for drawing various graphs to intuitively display data and results.

randomForest: Specifically used for training Random Forest models.

kernlab: Supports training Support Vector Machine (SVM) models.

pROC: Used for calculating and plotting ROC curves, extracting AUC values to quantify the model's discriminative ability.

glmnet: Supports the training of LASSO Regression models.

nnet: Used for training Neural Network models.

e1071: Provides training functions for Logistic Regression (LR) models.

tidyr: Used for data processing and format conversion, ensuring data meets analysis requirements.

**4.2 Model training and evaluation process.** *4.2.1 Data Preprocessing*: Firstly, use tidyr and other related tools to perform necessary cleaning and format conversion on differential gene expression data, including handling missing values, data normalization, etc., to ensure data quality and analysis accuracy.

*4.2.2 Model Training*: Random Forest (RF): Trained using the caret and randomForest packages.

LASSO Regression: Trained using the glmnet package, selecting the best regularization parameter through cross-validation.

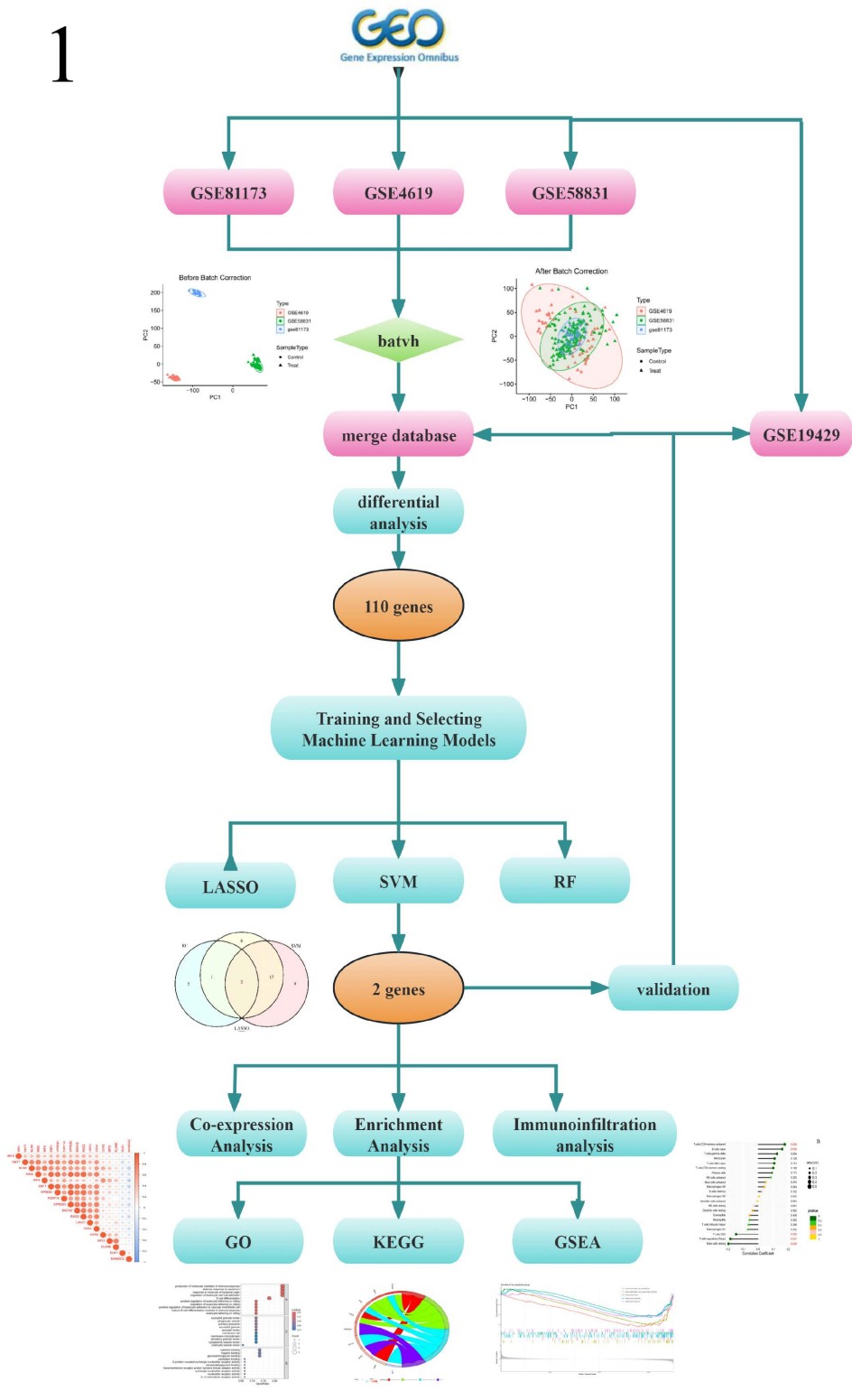

**Fig 1. Flow chart of research design and analysis.**

Support Vector Machine (SVM): Trained using the kernlab package, adjusting kernel functions and regularization parameters to optimize performance.

Neural Network (NN): Trained using the nnet package, adjusting network structure and hyperparameters to improve prediction accuracy.

Logistic Regression (LR): Trained with caret and regularized using glmnet, with model complexity and overfitting controlled by adjusting alpha and lambda.

Gradient Boosting Machine (GBM): Trained by calling the GBM algorithm through the caret package, adjusting parameters such as the number of trees, depth, and learning rate.

*4.2.3 Model Evaluation*: Use the DALEX package to interpret and evaluate the performance of each trained model.

Obtain key performance indicators such as accuracy, recall, F1 score, specificity, and AUC values through the model_performance function.

Use the pROC package to calculate and plot ROC curves of each model, intuitively displaying the model's discriminative ability.

Use the confusionMatrix function of the caret package to further analyze the classification performance of the models.

*4.2.4 Summary and Discussion of Results*: Finally, summarize all model performance indicators into a data frame, including model name, accuracy, recall, specificity, F1 score, and AUC value, etc. We will discuss these performance indicators, analyze the advantages and disadvantages of different models, and explain why the current model was chosen as the final analysis tool. In addition, we will also use the ggplot2 package to visually display these performance indicators for intuitive comparison of different models' performances.

**4.3 Model selection discussion.** We selected RF, LASSO, SVM, NN, LR, and GBM as core prediction models, mainly based on their universality and recognition, ability to handle high-dimensional data and nonlinear relationships, and expected performance on our specific dataset. RF and GBM can handle complex nonlinear relationships and high-dimensional data; LASSO achieves feature selection through regularization, suitable for datasets with a large number of features; SVM performs well in handling small sample data and nonlinear classification problems; NN, although requiring a large amount of data, has strong nonlinear modeling capabilities, making it potentially powerful in certain situations; LR is particularly well-suited for binary classification problems, featuring a simple implementation, high computational efficiency, and strong interpretability. Although other models, such as decision trees and naive Bayes, may also excel in certain situations, we have prioritized the selected models for their demonstrated stability and reliability in tackling the intricate non-linear relationships and high-dimensional features characteristic of gene expression data. We believe these models are better suited to meet our research needs.

## 5. Screening for disease-feature-related genes

In this study, we used three machine learning algorithms: Random Forest (RF), Lasso Regression, and Support Vector Machine (SVM) to screen for genes related to disease features. To ensure the accuracy, reliability, and robustness of the screening results, we decided to take the intersection of genes identified by these three algorithms as the final research subjects.

**5.1 Algorithm selection and implementation.** Random Forest (RF): We implemented the Random Forest algorithm using the "randomForest" package in R. This algorithm improves the accuracy of classification or regression by building multiple decision trees and aggregating their prediction results. We ranked the genes based on their importance scores in the Random Forest model and selected genes with higher scores as candidate genes.

Lasso Regression: We used the Lasso Regression algorithm from the "glmnet" package in R to identify genes related to disease features. Lasso Regression introduces an L1 regularization

term to achieve feature selection, capable of handling high-dimensional data and reducing overfitting. We adjusted the regularization parameter to select the most predictive genes.

Support Vector Machine (SVM): We implemented the SVM algorithm using the "e1071" package in R. SVM is a supervised learning algorithm based on the principle of structural risk minimization, adept at handling high-dimensional data and complex classification problems. We optimized model performance by adjusting SVM parameters and kernel function types and screened out genes strongly related to disease features.

**5.2  Intersection gene screening.**  After completing the analysis with the above three algorithms, we adopted an intersection screening strategy. Firstly, this is because numerous similar studies have used this method and proven its effectiveness in improving the accuracy and reliability of screening results. Secondly, different algorithms may have their own advantages and limitations when screening genes, and by taking the intersection, we can complement these shortcomings, thereby screening out more robust and reliable disease feature-related genes. Specifically, we selected those genes identified as related to disease features by all algorithms as the final research subjects. This method not only enhances the robustness of the screening results but also provides a more accurate and reliable gene set for subsequent disease research and diagnosis.

## 6.  Disease signature gene validation in the training set

We re-introduced the identified disease signature genes, IRF4 and ELANE, into the batch-corrected and consolidated training set of gene expression data. Utilizing the stat_compare_means function from the ggpubr package in R, we rigorously validated the differential expression of these two genes between the control and experimental groups, with a predefined significance threshold of $p < 0.001$. Following this, we employed the pROC package in R to plot ROC curves, enabling a quantitative assessment of the potential value of these genes as disease biomarkers within the training set.

## 7.  Validation of disease signature genes in the validation set

To validate the findings from the training set, we further incorporated the standardized GSE19429 validation set of gene expression data. Utilizing the stat_compare_means function from the ggpubr package in R once again, we re-verified the differential expression of IRF4 and ELANE genes between the control and experimental groups, maintaining the significance threshold at $p < 0.001$. Following this validation step, we employed the pROC package in R to plot ROC curves, allowing for a second round of quantitative assessment of the potential value of these two genes as disease biomarkers within the validation set.

## 8.  Differential analysis and co-expression analysis of disease signature gene groups

Based on the disease signature genes IRF4 and ELANE, we leveraged R to stratify the batch-corrected and consolidated training set of gene expression data into two distinct groups. These groups were defined by the expression levels of IRF4 and ELANE, with one group representing high expression (samples with elevated levels of both IRF4 and ELANE) and the other representing low expression (samples with relatively lower levels).After grouping, we utilized the limma package in R to conduct a detailed differential expression analysis of the remaining genes between these two groups. This step aimed to identify genes that showed significant changes in their expression patterns between the high and low expression groups, potentially revealing additional genes associated with the disease characteristics under investigation. Subsequently, we employed the corrplot package in R to perform a gene co-expression analysis.

### 9. Enrichment analysis

For the identified disease-specific genes IRF4 and ELANE, along with their closely related co-expressed genes, we utilized the clusterProfiler package in R to conduct Gene Ontology (GO) analysis and Kyoto Encyclopedia of Genes and Genomes (KEGG) pathway enrichment analysis. Additionally, we employed the GSEA (Gene Set Enrichment Analysis) function within the clusterProfiler package to further analyze gene sets. During the analysis, we set the significance level at a P-value less than 0.05, a criterion used to filter out statistically significant biological processes, cellular components, molecular functions, biological pathways, and disease-related pathways that are enriched in either the high or low expression groups defined by IRF4 and ELANE.

### 10. Immune-related functional analysis of disease-specific genes

We employed the "GSVA" package in R to perform Single-Sample Gene Set Enrichment Analysis (ssGSEA) on the disease-specific genes. This analysis aimed to assess the enrichment levels of immune-related gene sets, providing insights into the immune-related functionalities associated with the identified disease-specific genes.

### 11. Immunocell infiltration and immune cell correlation analysis of disease-specific genes

We utilized the CIBERSORT package in R, along with the corresponding CIBERSORT algorithm, to conduct an immunocell infiltration analysis on the batch-corrected and combined gene expression dataset from the training group. To ensure statistical significance in our results, we set the significance level at a P-value less than 0.05 and applied this criterion to filter out immune cell types that exhibited significant differences between the experimental and control groups. Subsequently, to delve deeper into the potential associations between disease-specific genes and immune cell abundances, we employed the cor.test function to perform Pearson correlation analysis. Using the same significance threshold of P-value less than 0.05, we successfully identified immune cell types that demonstrated significant correlations with the expression of disease-specific genes. This analysis provided valuable insights into the interplay between disease-related genetic signatures and the immune system's cellular components.

### 12. Statistical methods, software, and tools

All statistical analyses were conducted within the R environment. A two-sample t-test analyzed gene expression differences between groups. The Benjamini-Hochberg method was applied for multiple testing correction to control the false discovery rate (FDR). R language (version 4.4.1) was utilized for data processing and statistical analysis.

## II. Results

### 1. Data acquisition and processing

After downloading the platform description files and series matrix files for the four datasets, we conducted preliminary organization and merging. Subsequently, we obtained gene expression matrix files for each of the four datasets. Specifically: For the GSE81173 dataset, we generated a gene expression matrix containing 19,461 gene expression data points. For the GSE4619 dataset, we obtained a gene expression matrix with 22,880 gene expression data points. The GSE58831 dataset yielded a gene expression matrix also comprising 22,880 gene expression data points. Lastly, the GSE19429 dataset provided a gene expression matrix with 22,880 gene expression data points.

## 2. Identification of Differentially Expressed Genes (DEGs)

Through differential expression analysis of the gene expression datasets, we identified 370 DEGs in the GSE81173 dataset, 179 DEGs in the GSE58831 dataset, 46 DEGs in the GSE4619 dataset, and 68 DEGs in the GSE19429 dataset. (Fig 2A–2D)

## 3. Batch correction and differential analysis

Following the standardization process, we initially integrated the gene expression datasets from GSE81173, GSE58831, and GSE4619. Subsequently, we eliminated batch effects from the combined dataset. Then, we performed differential expression analysis on the batch-corrected and combined training set gene expression dataset, resulting in the identification of 110 differentially expressed genes along with their expression levels (Fig 3A-3C).

## 4. Training and selection of commonly used predictive models

We trained six models, including RF, LASSO, SVM, NN, LR, and GBM, and systematically evaluated their performances using a variety of evaluation metrics such as accuracy, recall, F1 score, specificity, and AUC value. Although all models performed similarly on most metrics, RF, NN, and GBM demonstrated excellent performance across all evaluated aspects. Particularly in terms of AUC value, RF, NN, and GBM all reached or approached 1.000, while SVM was 0.997, LASSO was 0.996, and KNN also performed well with an AUC value of 0.998 (Fig 4A–4E).

However, after an in-depth analysis of model applicability and data characteristics, we made the following choices:

Reason for not choosing NN (Neural Network): Despite its outstanding performance in the evaluation, NN has a high model complexity and typically requires a vast amount of training data to achieve ideal performance. In our study, there were 226 experimental groups and only 34 control groups, with a significant gap between the groups and a relatively small sample size, which may be insufficient to fully train the NN model, limiting its generalization ability. Furthermore, the parameter tuning process for NN is cumbersome and requires substantial computational resources, making it potentially not the most economically efficient choice for this study.

Reason for not choosing LR (Logistic Regression): While LR is a simple model with high computational efficiency and stable performance on small sample data, it may struggle with handling nonlinear relationships and high-dimensional data. Our differential gene expression data may contain complex nonlinear relationships and high feature dimensions, making LR potentially inadequate in capturing underlying patterns in the data.

Reason for not choosing GBM (Gradient Boosting Machine): GBM is a powerful ensemble learning method capable of handling nonlinear relationships and high-dimensional data. However, in our study, due to the large difference in sample size between the experimental and control groups, GBM may have difficulty balancing the importance of different class samples during training, leading to model bias. Additionally, GBM models typically contain numerous parameters, making the tuning process complex and requiring substantial computational resources.

Conversely, the three models of RF (Random Forest), LASSO (Lasso Regression), and SVM (Support Vector Machine) are highly recognized in the field of bioinformatics, with solid theoretical foundations and mature implementation methods [18–20]. They can handle nonlinear relationships, high-dimensional data, and imbalanced sample sizes, with good model interpretability. Given our data characteristics, these three models performed well and were relatively robust. Therefore, after comprehensive consideration, we decided to include the three machine learning algorithms of RF, LASSO, and SVM for further research.

## 5. Screening of disease-specific genes

We applied Lasso regression, Random Forest algorithm, and Support Vector Machine (SVM) algorithm to a gene pool containing 110 differentially expressed genes to screen for candidate disease-specific genes. Specifically, Lasso regression identified 25 candidate genes from the gene pool. The Random Forest algorithm further screened out 8 candidate genes. The SVM algorithm also independently selected 8 candidate genes. To achieve higher accuracy and reliability in identifying disease-specific genes, we cross-validated the gene sets selected by these algorithms. After careful analysis and comparison, we ultimately identified two highly

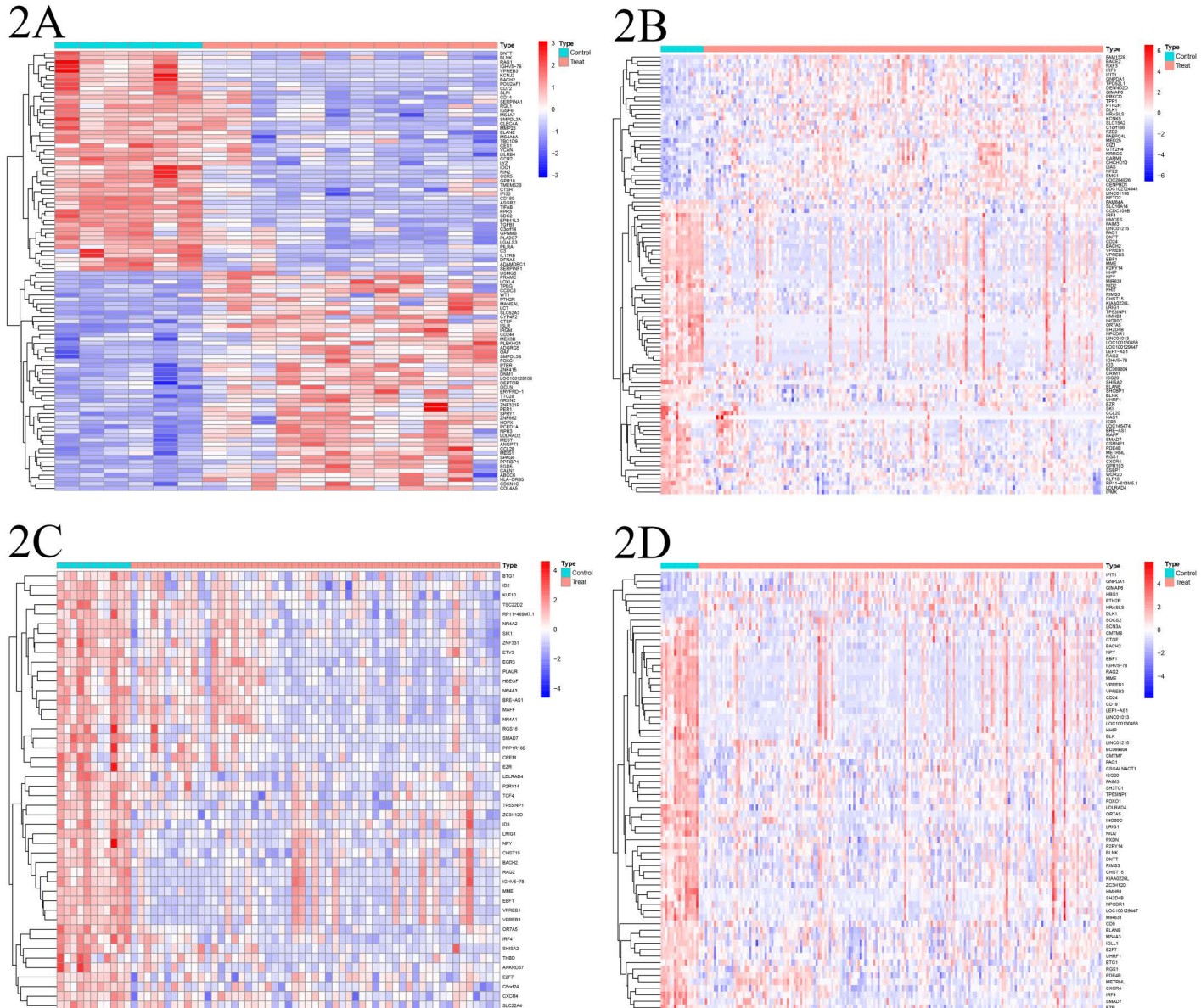

**Fig 2. A. Displays a heatmap of differentially expressed genes in the GSE81173.** B. Shows a heatmap for the GSE58831. C. Illustrates differentially expressed genes in the GSE4619. D. A thorough depiction of the heatmap for the validation group GSE19429. In these heatmaps, red indicates upregulated genes, and blue represents downregulated genes. "Control" denotes the normal control group, while "Treat" refers to MDS.

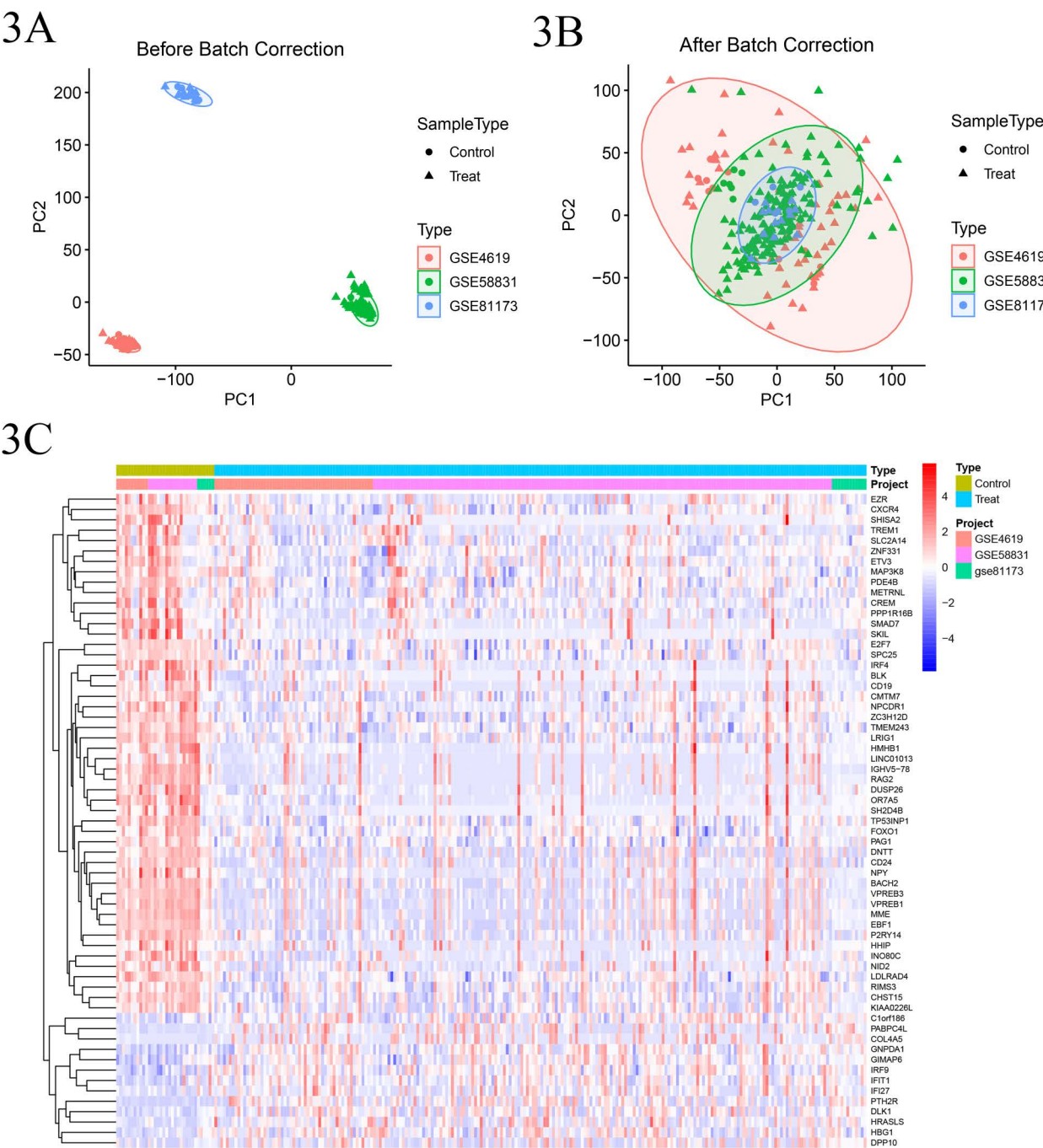

**Fig 3. A. Scatter plot of the training set data before batch correction; B: Scatter plot of the training set data after batch correction.** Dots (●) represent control group data; Triangles (▲) represent experimental group data; Color code: Red for GSE4619 dataset, Green for GSE58831 dataset, Blue for GSE81173 dataset. C. Heatmap of the combined training set data after batch effect removal, where yellow-green represents the normal control group, blue represents the disease group, orange, pink, and green distinguish different datasets included, red indicates high expression, and blue indicates low expression.

correlated and reliable disease-specific genes: IRF4 and ELANE. These two genes showed significant correlation in all three algorithms and were therefore considered strong candidates for disease-specific genes (Fig 5A–5F).

## 6. Validation of disease-specific genes in the training set

Initially, we conducted a validation analysis within the training set dataset, using a p-value < 0.001 as the criterion for statistical significance. The results indicated that there were significant differences in the gene expression levels of IRF4 and ELANE between the experimental group and the normal control group within the training set gene expression dataset. Specifically, both IRF4 and ELANE were found to be downregulated in the experimental group compared to their upregulated expression in the control group. Subsequently, based on the training set dataset, we constructed ROC curves for the disease-specific genes IRF4 and ELANE, yielding an AUC value of 0.929 for IRF4 and 0.799 for ELANE. (Fig 6A–6B)

## 7. Independent validation using an independent dataset

To further validate our findings, we included the independent dataset GSE19429 for analysis, applying a p-value < 0.001 as the threshold for statistical significance. The results showed that there were significant differences in the gene expression levels of IRF4 and ELANE between the experimental group and the normal control group within the independent dataset as well. Specifically, both IRF4 and ELANE were downregulated in the experimental group and upregulated in the control group. Subsequently, based on the independent dataset GSE19429, ROC

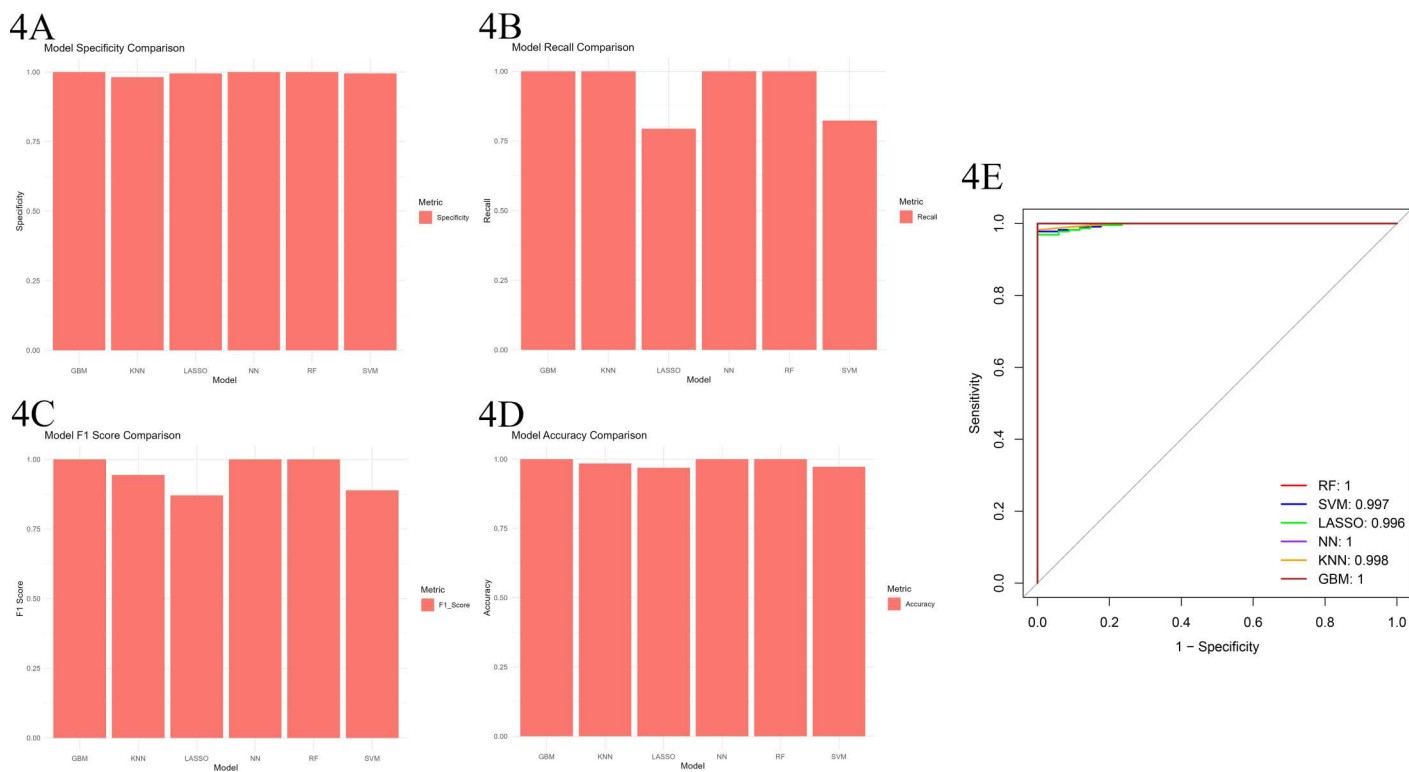

**Fig 4. A. Bar chart for accuracy evaluation of six models.** B. Bar chart for recall evaluation of six models. C. Bar chart for F1 score evaluation of six models. D. Bar chart for specificity evaluation of six models. E. The ROC curve chart demonstrates the classification performance of six machine algorithms in the differential gene analysis task. The AUC value quantitatively reflects the overall classification capability of each model.

curves were constructed for the disease-specific genes IRF4 and ELANE, yielding an AUC value of 0.938 for IRF4 and 0.791 for ELANE (Fig 7A–7D).

## 8. Differential analysis and co-expression analysis of disease-specific gene groups

First, based on the expression levels of IRF4 and ELANE, we utilized R language to divide the batch-corrected training set gene expression dataset into two groups: a high-expression group and a low-expression group. The high-expression group represented samples with higher expression levels of IRF4 and ELANE, while the low-expression group represented samples with lower expression levels. In the IRF4-based grouping, we observed negative correlations between the expression of IRF4 and DLK1, MAMDC2, and positive correlations with 15 genes including LRIG1, P2RY14, DNTT, CD24, among others. Similarly, in the ELANE-based grouping, WT1 exhibited a negative correlation with ELANE expression, while 47 genes such as CFD, HAL, CLEC12A, NKG7, showed positive correlations with ELANE expression. Subsequently, we performed co-expression analysis on the gene sets significantly associated with IRF4 and ELANE using the corrplot package. The results showed that 17 genes including DNTT, BLNK, MME, exhibited significant co-expression relationships with IRF4. Additionally, DLK1 and MAMDC2 displayed a positive correlation with each other and negative correlations with the remaining 15 genes to varying degrees. For ELANE, we found 20 genes including PRTN3, AZU1, MPO, to have significant co-expression relationships. Except for WT1, these genes exhibited varying degrees of positive correlations with each other (Fig 8A–8D).

## 9. Enrichment analysis

Utilizing Gene Ontology (GO) and Kyoto Encyclopedia of Genes and Genomes (KEGG), we conducted an enrichment analysis on the identified disease-characteristic-related genes IRF4 and ELANE, along with their co-expressed genes. A significance threshold of $P < 0.05$ was employed in this analysis.

**9.1. Enrichment analysis of IRF4.** The IRF4 gene was significantly enriched in multiple biological processes (BPs), including the production of molecular mediators of immune response, bacterial defense response, response to molecules of bacterial origin, and regulation of leukocyte-cell adhesion. At the cellular component (CC) level, it was primarily enriched in phagocytic vesicle-related components. In terms of molecular function (MF), functions related to cytokine binding were significantly enriched. KEGG pathway analysis revealed significant enrichment of IRF4 in pathways such as hematopoietic cell lineage and primary immunodeficiency (Fig 9A–9B).

**9.2 Enrichment analysis of ELANE.** The ELANE gene was primarily enriched in BPs associated with humoral immune response, bacterial defense response, and regulation of chemotaxis. At the CC level, these genes were mainly enriched in cytoplasmic vesicle lumen and secretory granule lumen-related components. In terms of MF, functions related to endopeptidase activity were significantly enriched. KEGG pathway analysis showed significant enrichment of ELANE in pathways such as neutrophil extracellular trap formation and transcriptional misregulation in cancer (Fig 9C–9D).

**9.3 Enrichment analysis of Co-expressed genes.** Using chord diagram visualization, we demonstrated the co-enrichment of co-expressed genes closely associated with IRF4 and ELANE across various GO and KEGG categories. Co-expressed genes strongly linked to IRF4 were primarily enriched in GO categories such as leukocyte tethering or rolling, immune response, bacterial defense response, and B cell differentiation. Meanwhile, co-expressed

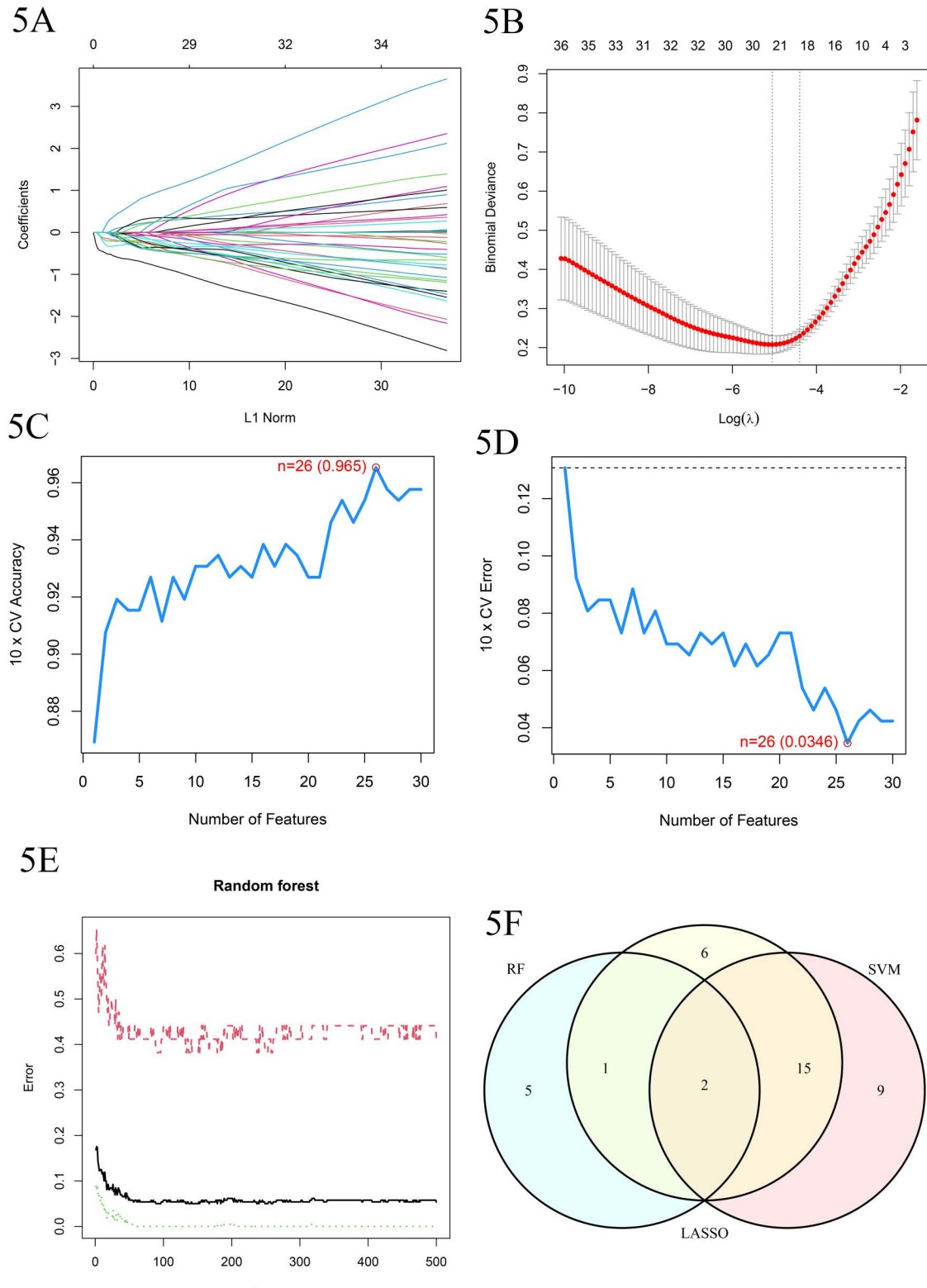

**Fig 5. Screening of variables based on Lasso regression.** A. The variation characteristics of the coefficient of variables; B. The selection process of the optimum value of the parameter λ in the Lasso regression model by cross-validation method. C. Bubble chart of gene importance related to diseases obtained by the Random Forest algorithm. D. The error rate plot related to diseases obtained by the Support Vector Machine (SVM) algorithm, where the x-axis represents the size of the feature subset, and the y-axis

represents the corresponding error rate. E. The accuracy plot related to diseases obtained by the Support Vector Machine (SVM) algorithm, where the x-axis represents the size of the feature subset, and the y-axis represents the corresponding accuracy rate. F. Through comparative analysis of three algorithms—Lasso regression, Support Vector Machine (SVM), and Random Forest (RF)— we have obtained a schematic diagram of disease-related genes that are commonly identified by these methods.

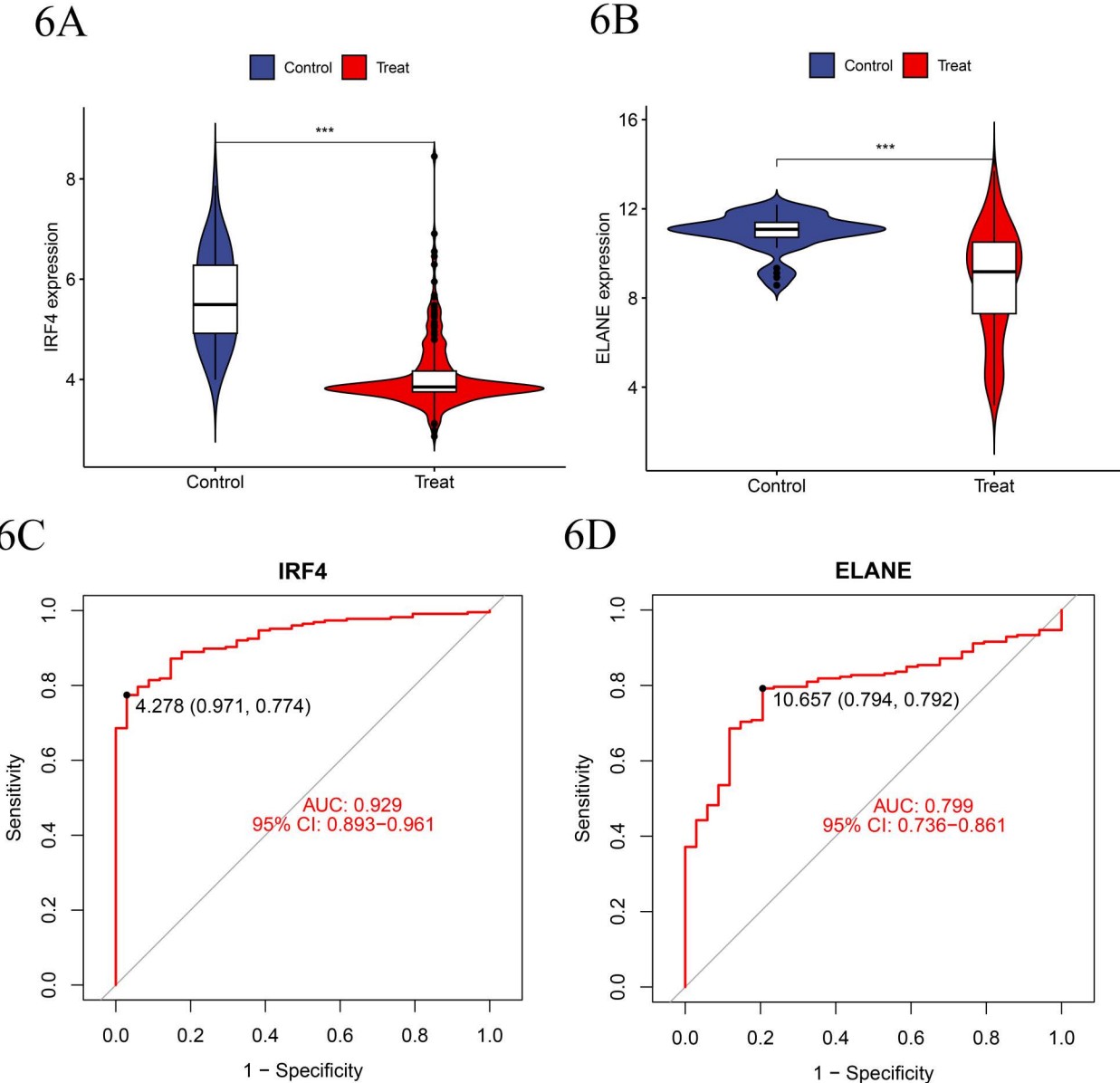

**Fig 6. A-B. Expression of IRF4 and ELANE in the experimental group compared to the normal control group showed significantly reduced expression levels of IRF4 and ELANE in the MDS disease group, highlighting their potential role in the pathogenesis of these genes.** C-D. Based on the merged training group dataset after batch effect removal, construct ROC curves for disease-related genes IRF4 and ELANE.

genes closely related to ELANE were primarily enriched in GO categories including humoral immune response, antibacterial humoral response, and defense response to fungi and other organisms (Fig 10A–10D).

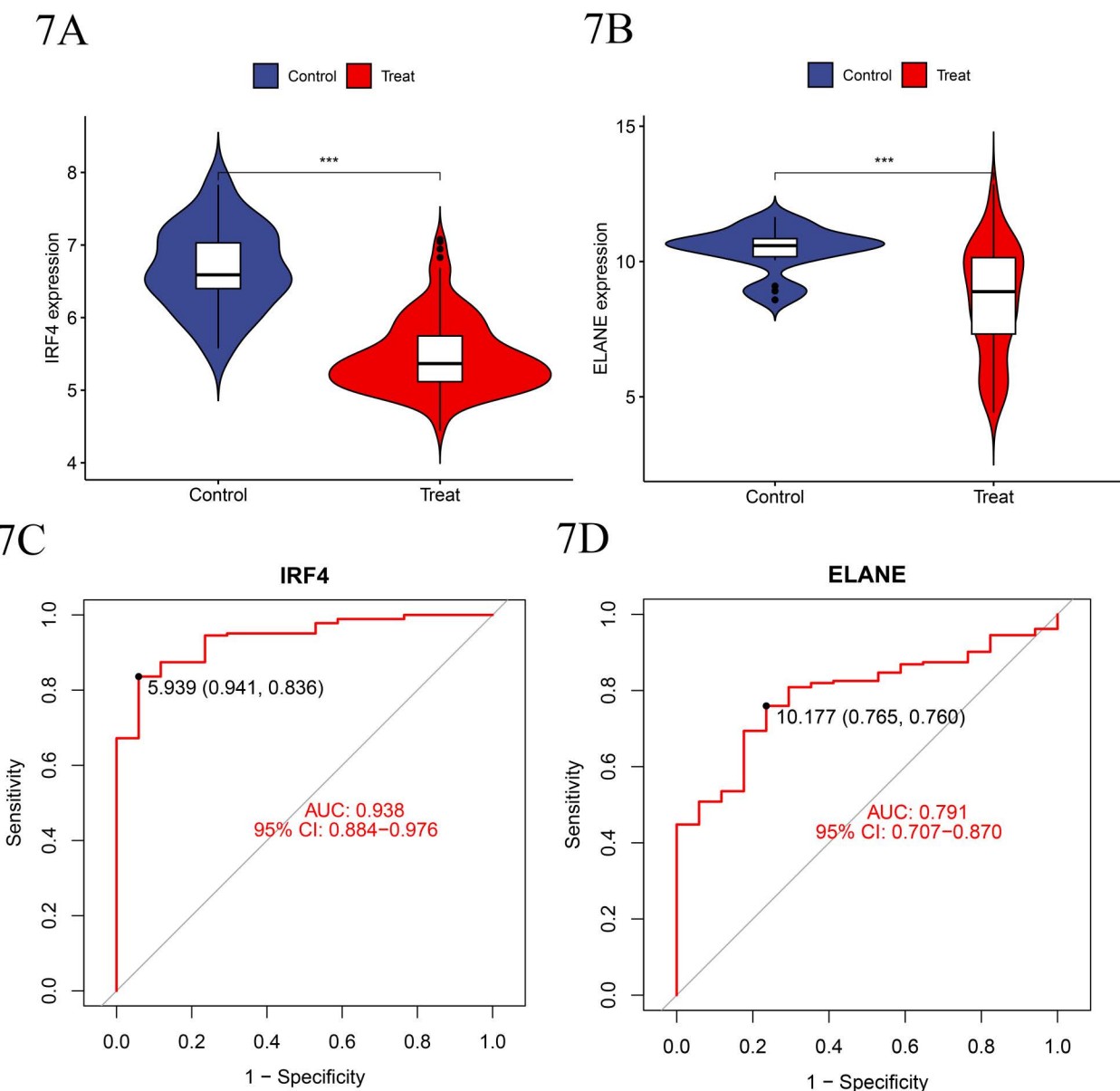

**Fig 7. A-B. Expression of IRF4 and ELANE in MDS - Indicates that compared to the normal control group, the expression levels of both in the MDS disease group show significant reductions at different levels (P < 0.001), suggesting that the reduction in their activity may affect the mechanism or progression of the disease.** C-D. The ROC curve illustrates the significance of genes related to MDS. In the ROC curve analysis, the significant AUC of the IRF4 gene is 0.938, and the significant AUC of the ELANE gene is 0.791.

**9.4 Gene set enrichment analysis.** In the high-expression group of IRF4, biological processes like DNA replication, chromosome segregation, DNA-templated DNA replication, mitotic sister chromatid segregation, and regulation of chromosome segregation, along with KEGG pathways such as cell cycle, DNA replication, oocyte meiosis, primary immunodeficiency, and proteasome, were significantly enriched. In contrast, the low-expression group showed enrichment in biological processes like hemostasis, platelet activation, regulation of fluid level, and wound healing, along with related KEGG pathways like arachidonic acid metabolism, complement and coagulation cascades, endocytosis,

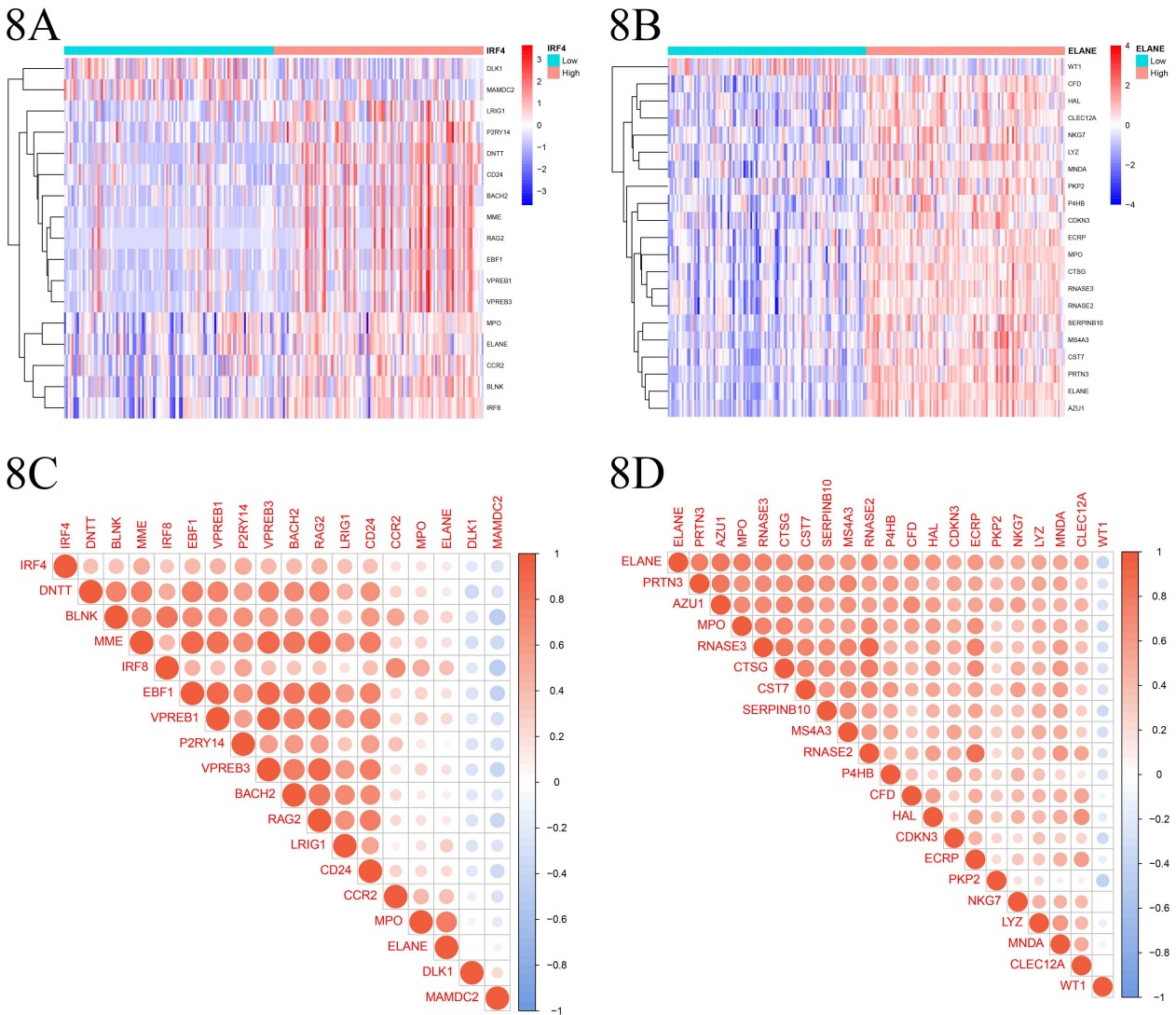

**Fig 8. A-B. Differential expression heatmap of IRF4 and ELANE genes, grouped by high and low expression of disease-associated features.** Red represents positive correlation, blue represents negative correlation. Left panel: IRF4, Right panel: ELANE. "HIGH" and "LOW" indicate expression levels of disease-associated genes. C-D. Correlation Matrix Chart: Illustrating the correlation between disease-related genes and target genes. Blue tiles signify negative regulation, whereas red tiles indicate positive regulation. The intensity of the color reflects the strength of the correlation.

focal adhesion, and tight junction. For the high-expression group of ELANE, biological processes such as antibacterial humoral response, antimicrobial humoral immune response mediated by antimicrobial peptides, defense response to Gram-negative bacterium, and related KEGG pathways like cell cycle, DNA replication, hematopoietic cell lineage, oocyte meiosis, and systemic lupus erythematosus were significantly enriched. In the low-expression group, biological processes like cardiac cell fate determination, endosomal transport, and autophagy, along with molecular functions like ubiquitin-like protein ligase activity and transferase activity, were enriched. Additionally, multiple KEGG pathways including epithelial cell signaling in Helicobacter pylori infection, Hedgehog signaling pathway, long-term potentiation, maturity-onset diabetes of the young, and phosphatidylinositol signaling system were significantly enriched (Fig 11A–11H).

## 10. Immunological function analysis

The influence of IRF4 and ELANE gene expression levels on immune-related functions was evaluated through single-sample gene set enrichment analysis (ssGSEA). Our analysis revealed significant differences in the scoring distributions across immune-related gene sets, with respect to IRF4 gene expression (P ≤ 0.001). Specifically, B cell-associated immune functions were markedly enhanced in the low-IRF4 expression group, whereas cell lytic activity and functions related to plasmacytoid dendritic cells were significantly elevated in the high-IRF4 expression group. Analogously, ELANE gene expression also demonstrated a profound impact (P < 0.001), with marked augmentation observed in mast cell, neutrophil, and plasmacytoid dendritic cell-related immune functions within the high-ELANE expression cohort (Fig 12A–12B).

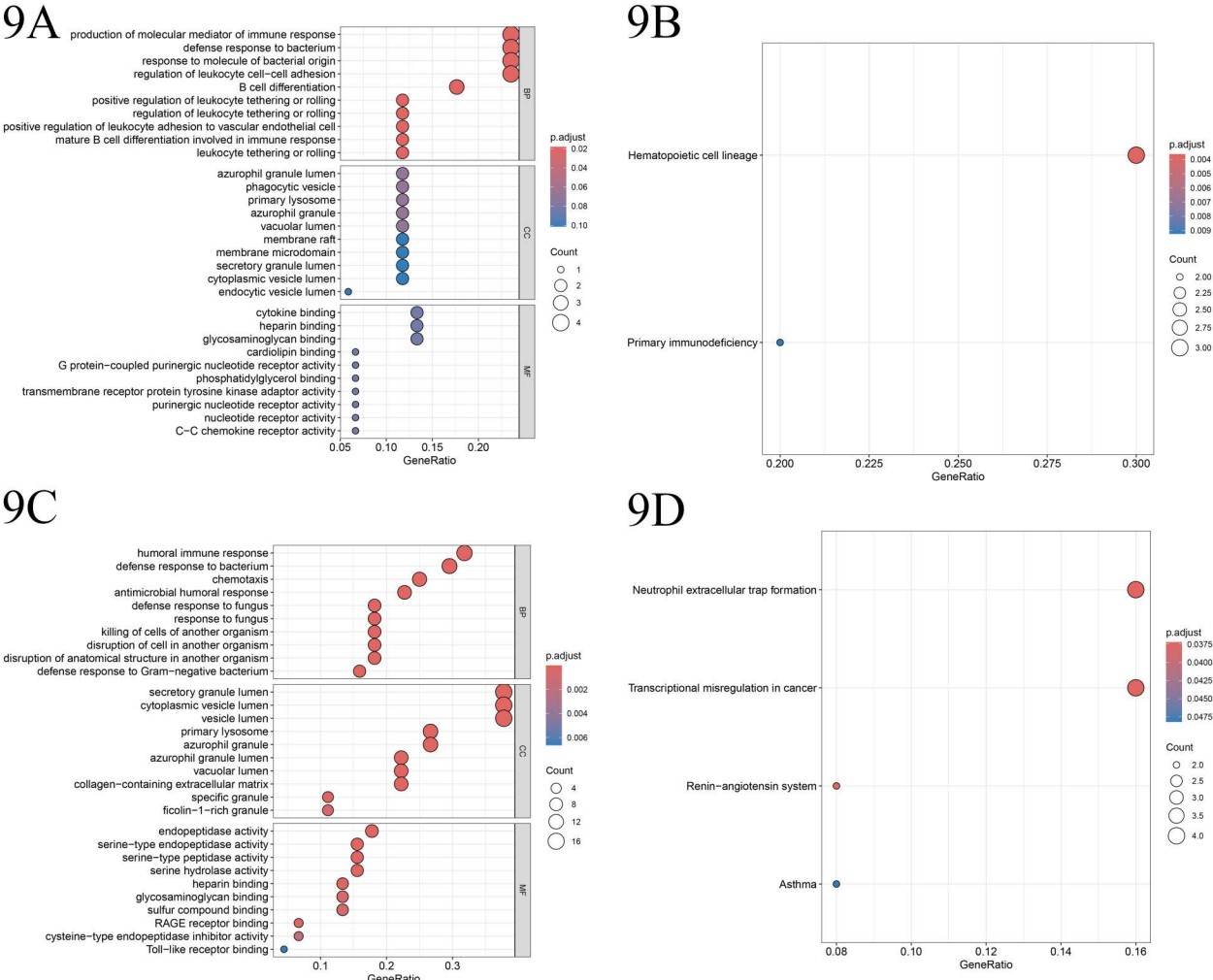

**Fig 9. A. Illustrates the GO enrichment analysis pathway diagram for gene IRF4, where the x-axis denotes the count of enriched genes, and the y-axis represents the enrichment significance.** B. Depicts the KEGG enrichment analysis pathway diagram for gene IRF4, with the x-axis indicating the number of enriched genes and the y-axis signifying the enrichment significance. C. Displays an alternative aspect of the GO enrichment analysis pathway diagram for gene ELANE, wherein the x-axis specifies the gene ratio, and the y-axis depicts the enrichment significance. D. Showcases the KEGG enrichment analysis pathway diagram for gene ELANE, with the x-axis indicating the gene ratio and the y-axis denoting the enrichment significance.

## 11. Immunocellular infiltration and correlation analysis of disease-associated genes with immune cell types

Based on the gene expression dataset from the batch-corrected training group, we conducted an immune cell infiltration analysis and objectively presented the enrichment differences in the expression levels of immune-related functions between the experimental and control groups through box plots. Our analysis revealed statistically significant differences (P < 0.05) in the presence of CD4 memory activated T cells, resting dendritic cells, and activated mast cells between the two groups, suggesting altered expression or activity of these cell types in the experimental group compared to the control. Subsequently, Pearson correlation analysis uncovered notable correlations between the genes IRF4 and ELANE with various immune cell types. Specifically, IRF4 exhibited a positive correlation with CD4 memory T cells and naive B cells, while displaying a negative correlation with CD8 + T cells, regulatory T cells, and resting mast cells. Similarly, ELANE correlated positively with monocytes and negatively with naive CD4+ T cells and naive B cells. These findings contribute to a deeper understanding of the complex interplay between immune cell subsets and their regulatory genes in the context of the studied biological system (Fig 13A-13C).

## III. Discussion

MDS, a malignant clonal stem/progenitor cell disorder originating from CD34+ cells, primarily impacts individuals over 65 years old, with a global incidence rate ranging from 2 to 12 per 100,000 individuals [21]. Given the intensifying aging population, this ratio is projected to continue rising.1 To delve deeper into its pathogenesis, we have integrated bioinformatics and machine learning approaches, aiming to uncover novel potential therapeutic targets and strategies for clinical research and treatment of MDS. In this study, we zeroed in on the gene expression profile of bone marrow CD34+ cells in MDS, systematically analyzing four datasets encompassing both MDS patient and healthy control bone marrow samples. Through rigorous screening, we pinpointed two crucial disease-signature genes: IRF4 and ELANE. Both genes exhibited significantly lower expression levels in CD34+ cells from MDS patients, underscoring their pivotal roles in immune response and regulation of cellular differentiation, which are intimately linked to the initiation and progression of MDS. Our findings not only corroborate previous research but also reinforce the pivotal status of these genes in the pathogenesis of MDS, thereby offering valuable insights into potential therapeutic strategies for the future.

Previous independent studies have unequivocally demonstrated a significant downregulation of IRF4 gene expression in myelodysplastic syndromes (MDS). Specifically, the work by Vasikova A et al. not only unveiled the reduced expression of IRF4 across distinct genetic subsets of CD34+ cells in both early and advanced MDS patients, but also provided crucial insights into the pivotal role of IRF4 in the pathogenesis of MDS [22]. Concurrently, a plethora of research has highlighted a consistent downregulation of IRF4 expression across the myeloid disease spectrum, encompassing acute myeloid leukemia (AML), chronic myeloid leukemia (CML), and a range of hematopoietic cancer cell lines [23–27]. This cross-disease consensus underscores the central importance of IRF4 in hematological malignancies, suggesting that the downregulation of IRF4 in MDS, as a subset of myeloid disorders, may represent a shared critical biological feature among these diseases. Furthermore, animal model studies have offered more tangible evidence. In murine models, the absence of IRF4 has been proven to exacerbate the progression of myeloid leukemia [28], reinforcing the essential role of IRF4 in blood disorders and presenting potential avenues for disease intervention and treatment. Given the intimate relationship between MDS and these myeloid disorders, these

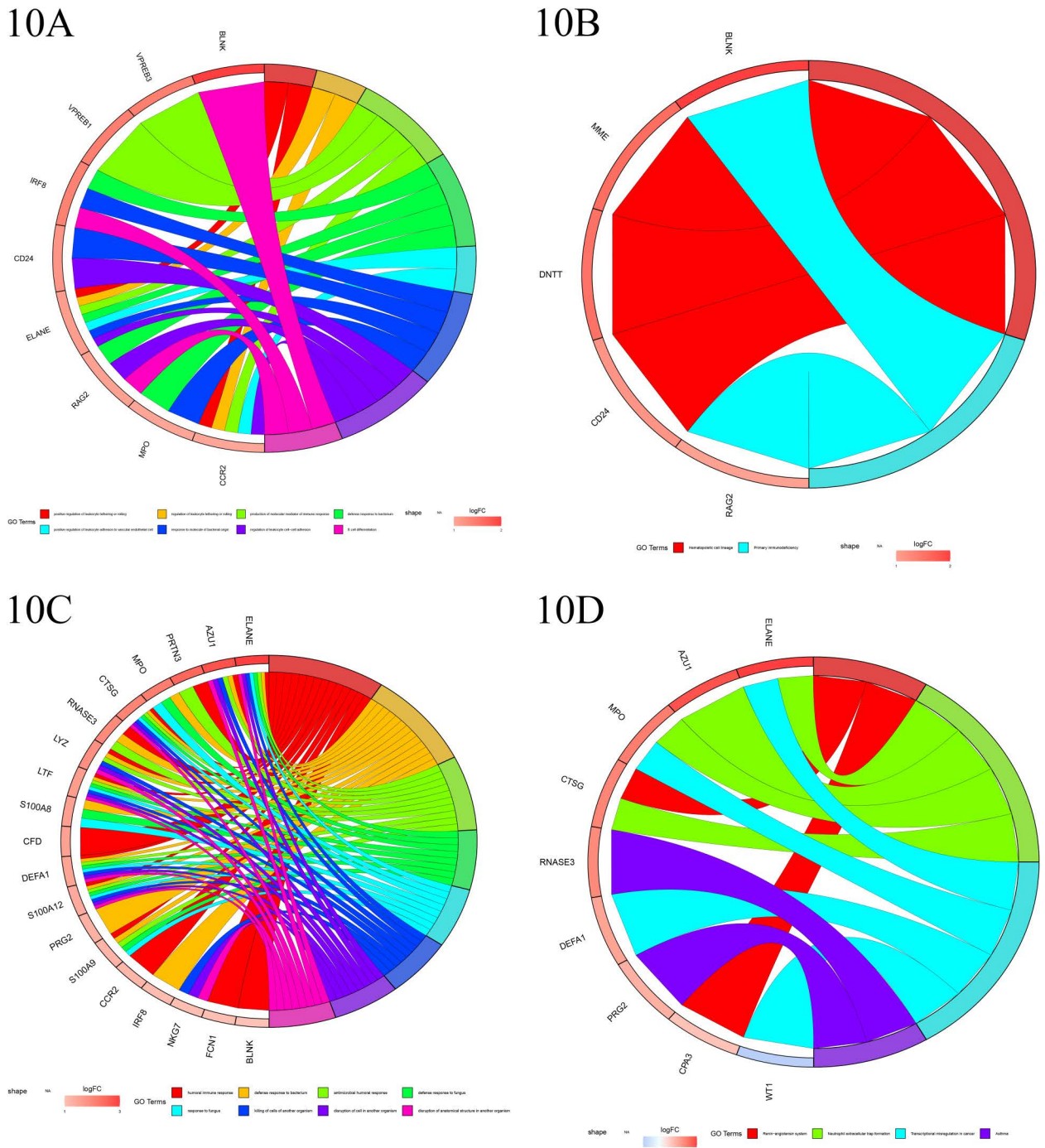

**Fig 10. A-B. The chord diagram illustrates the enrichment of IRF4 and its co-expressed genes across Gene Ontology (GO) categories (left panel) and Kyoto Encyclopedia of Genes and Genomes (KEGG) pathways (right panel).** The left side lists gene names, with colors differentiating the direction of expression (red for upregulation, blue for downregulation). The multicolored bars on the right represent distinct GO categories and KEGG pathways, while the thickness of the connecting lines reflects the number of co-enriched genes shared between them. **C-D.** The chord diagram displays the enrichment of ELANE and its co-expressed genes across Gene Ontology (GO) categories (left panel) and Kyoto Encyclopedia of Genes and Genomes (KEGG) pathways (right panel). The left side enumerates gene names, with colors distinguishing the direction of expression (red for upregulation, blue for downregulation). The multicolored bars on the right signify various GO categories and KEGG pathways, while the thickness of the connecting lines indicates the quantity of co-enriched genes shared among them.

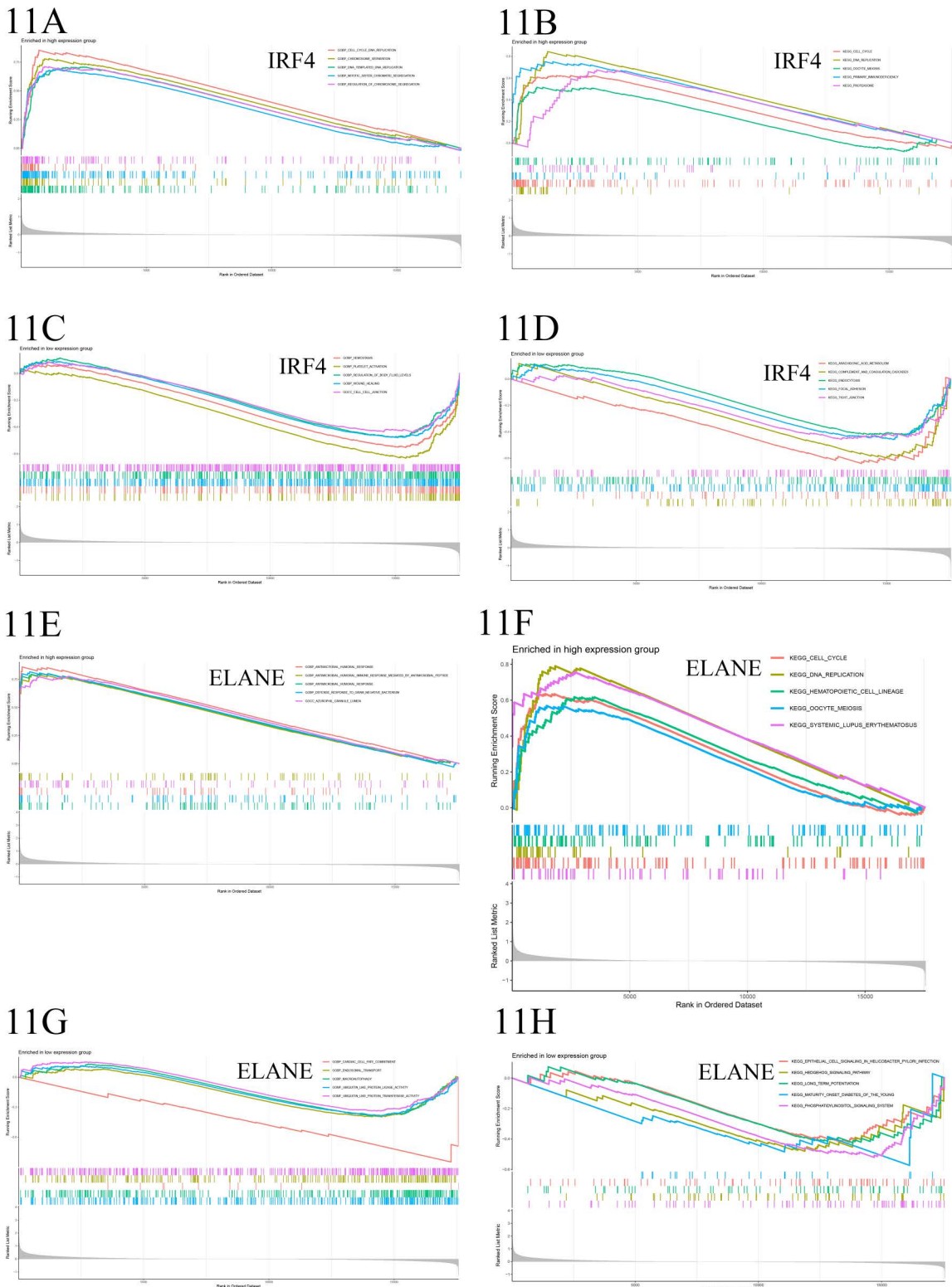

**Fig 11. A-B. This Enrichment Score (ES) plot displays the enrichment of IRF4 and its co-expressed gene high expression group across Gene Ontology (GO) categories and Kyoto Encyclopedia of Genes and Genomes (KEGG) pathways.** The horizontal axis represents sorted genes, the vertical axis indicates the enrichment score, and colored bars distinguish different GO categories and KEGG pathways. Peaks of the ES line towards the upper left indicate significant enrichment of these categories and pathways within the high expression group. **C-D.** This Enrichment Score (ES) plot exhibits the enrichment of IRF4 and its co-expressed gene low

expression group across Gene Ontology (GO) categories and Kyoto Encyclopedia of Genes and Genomes (KEGG) pathways. The horizontal axis represents sorted genes, the vertical axis indicates the enrichment score, and colored bars differentiate various GO categories and KEGG pathways. Peaks of the ES line towards the lower right signify significant enrichment of these categories and pathways within the low expression group. **E-F**. This Enrichment Score (ES) plot demonstrates the enrichment of ELANE and its co-expressed gene high expression group across Gene Ontology (GO) categories and Kyoto Encyclopedia of Genes and Genomes (KEGG) pathways. The horizontal axis represents sorted genes, the vertical axis indicates the enrichment score, and colored bars distinguish between different GO categories and KEGG pathways. Peaks of the ES line located towards the upper left signify significant enrichment of these categories and pathways within the high expression group. **G-H**. This Enrichment Score (ES) plot illustrates the enrichment of ELANE and its co-expressed gene low expression group across Gene Ontology (GO) categories and Kyoto Encyclopedia of Genes and Genomes (KEGG) pathways. The horizontal axis represents sorted genes, the vertical axis indicates the enrichment score, and colored bars differentiate various GO categories and KEGG pathways. Peaks of the ES line positioned towards the lower right indicate significant enrichment of these categories and pathways within the low expression group.

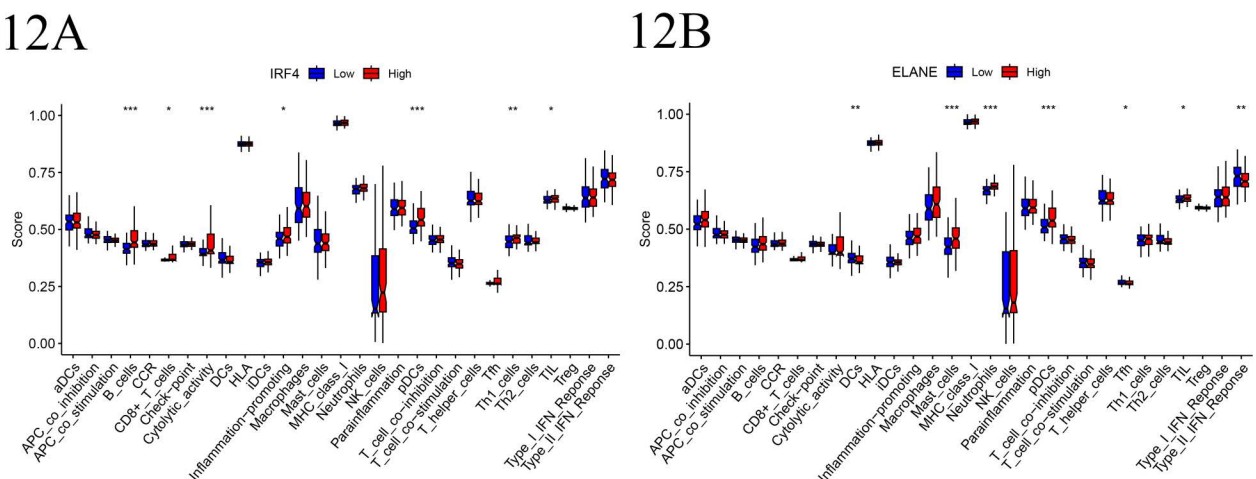

**Fig 12. A-B. The boxplots present the distribution of scores for IRF4 and ELANE immune-related functions in low-expression and high-expression groups.** The horizontal axis corresponds to the immune functions, and the vertical axis indicates the activity score. Blue and red colors signify the low-expression and high-expression groups, respectively. Significant differences are denoted by *** (p < 0.001), ** (p < 0.01), and * (p < 0.05).

findings hold significant implications for exploring therapeutic strategies for MDS. Collectively, these research achievements not only deepen our understanding of the role of IRF4 in MDS pathogenesis but also pave the way for the development of novel therapeutic approaches that target IRF4 dysfunction. The consistency of IRF4 downregulation across myeloid diseases underscores its potential as a universal therapeutic target, offering hope for more effective treatments that can span across multiple hematological malignancies.

In delving deeper into the pathogenic mechanisms of myelodysplastic syndromes (MDS), we have observed that the pivotal gene IRF4 plays a significant role across various hematological disorders. Prior studies have illuminated the regulatory pathways of IRF4 in diverse disease contexts and its intimate association with tumor progression—notably, the work by Lopez-Girona A et al. revealed that immunomodulatory drugs (IMiDs) exert their tumor-suppressive effects by reducing IRF4 activity or expression levels, and this effect is modulated by the expression level of cereblon (CRBN) [29]. This discovery offers clues into the potential mechanisms of IRF4 in MDS. We speculate that in MDS, IRF4 may participate in disease progression by influencing the expression or function of CRBN or other related proteins. Firstly, given that IRF4 is a crucial transcription factor, it plays a pivotal role in regulating cellular

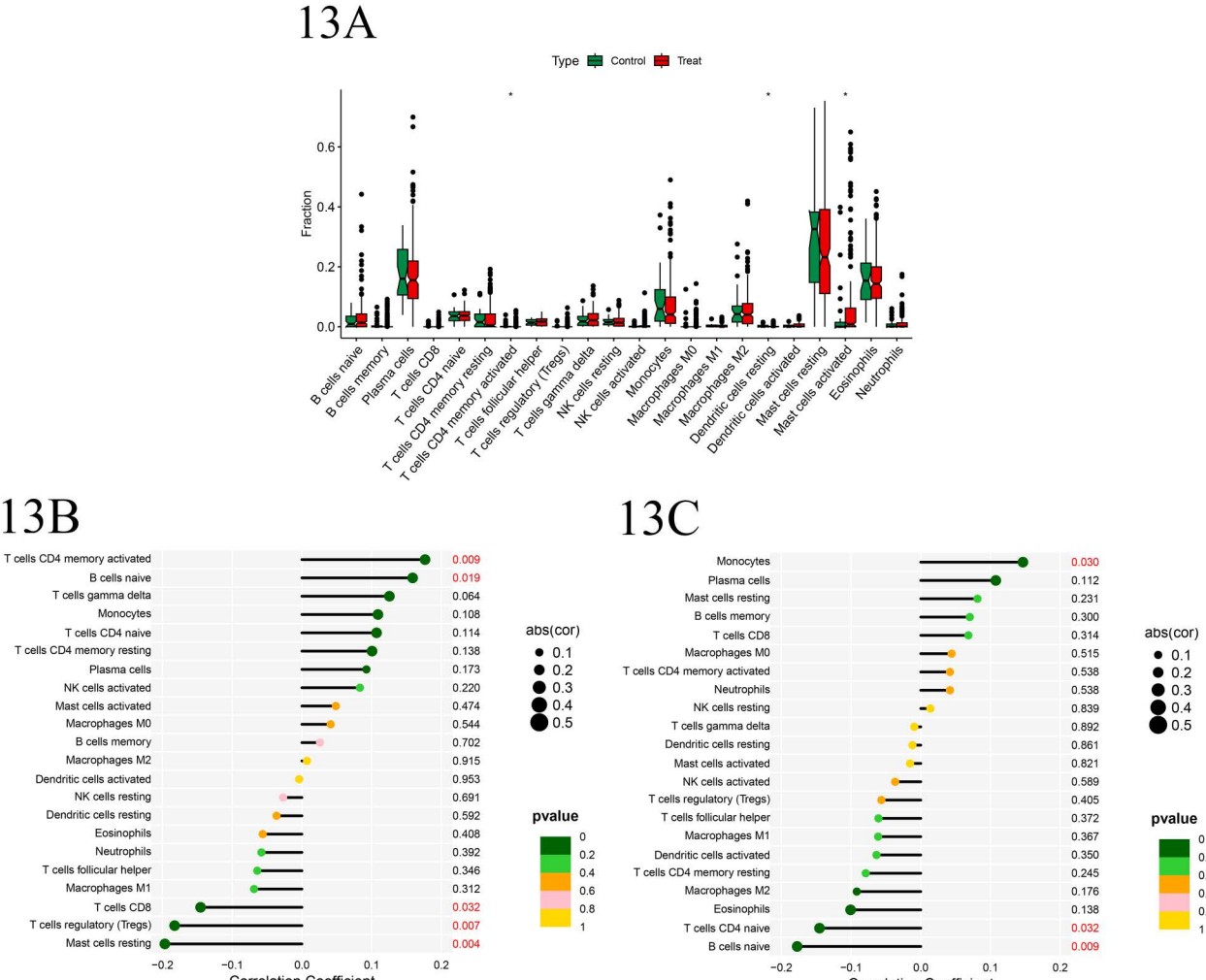

**Fig 13. A. The boxplot displays the distribution of enrichment scores for disease-related immune functions between the control group and the experimental group.** The horizontal axis represents the immune-related functions, while the vertical axis indicates the enrichment score. Green and red colors are used to distinguish the control group and the experimental group, respectively. Significant differences are marked with *** ($p < 0.001$), ** ($p < 0.01$), and * ($p < 0.05$), indicating varying degrees of statistical significance. **B-C.** bubble chart illustrating the correlation between IRF4 and ELANE genes with various immune cell types. The vertical axis represents immune cell types, while the horizontal axis depicts the Pearson correlation coefficient, with positive values indicating positive regulation and negative values indicating negative regulation. significant correlations ($P < 0.05$) are highlighted in red font.

processes such as proliferation, differentiation, and apoptosis [30–33]. In MDS, abnormalities in these biological processes are often intimately linked to disease initiation and progression. Based on this premise, we can hypothesize that IRF4 in MDS may impact disease progression by modulating these biological processes. Furthermore, the intricate interplay between IRF4 and its downstream targets, including but not limited to CRBN, may constitute a regulatory network that is dysregulated in MDS, contributing to the pathological features of the disease. Exploring this network and identifying key nodes for therapeutic intervention could lead to the development of novel strategies for managing MDS.

Secondly, regarding the ELANE gene, it encodes neutrophil elastase, an enzyme crucial for neutrophil function. Neutrophils play a pivotal role in the human immune system, combating pathogens and clearing necrotic cells through the release of various enzymes, including

neutrophil elastase. Previous studies have primarily focused on the relationship between ELANE gene mutations and severe congenital neutropenia (SCN) as well as their progression to myelodysplastic syndromes/acute myeloid leukemia (MDS/AML) [34,35]. These studies have shown that SCN can be caused by mutations in multiple genes, including ELANE, and that ELANE mutations are the most common genetic defect leading to the development of MDS/leukemia from SCN. For instance, Krutein et al. revealed that heterozygous mutations in ELANE encoding the potent serine protease neutrophil elastase (NE) cause cyclic neutropenia (CyN), which is the most common cause of severe congenital neutropenia (SCN) [36,37]. However, apart from mutations, our study discovered a significant reduction in ELANE gene expression levels in CD34+ cells from MDS patients. This novel finding provides a fresh perspective on understanding the pathogenesis of MDS. We speculate that the low expression of the ELANE gene may impair the normal functions of neutrophils, including their differentiation and bactericidal capabilities, thereby contributing to the pathogenesis of MDS. Prior research, such as the work by Nanua S et al., validated in an Elane-targeted mutation (G193X) transgenic mouse model that ELANE mutations lead to a block in neutrophil differentiation [36]. This finding further underscores the crucial role of the ELANE gene in the differentiation process of neutrophils. Additionally, studies by Cui et al. and Peng B et al. have highlighted the role of neutrophil elastase (ELANE) in killing cancer cells [38,39]. These studies demonstrate that catalytically active neutrophil elastase (ELANE) released by human neutrophils can selectively kill multiple cancer cell types while sparing non-cancerous and normal cells, significantly reducing tumor formation. This discovery emphasizes the potential value of the ELANE gene in anticancer processes. In conclusion, our study uncovered the low expression of IRF4 and ELANE genes in CD34+ cells from MDS patients and explored their potential roles in the pathogenesis of MDS. These findings not only offer new insights into MDS research but also provide valuable clues for future therapeutic strategies. Future research can further delve into the specific mechanisms underlying these gene expression changes and explore ways to modulate their expression to improve treatment outcomes for MDS patients. Lastly, the primary limitations of our study are as follows: the MDS-related gene networks identified through bioinformatics have yet to be experimentally validated in patients, and the significant heterogeneity of MDS has hindered the adequate identification of hub genes across its various subtypes. While our research has offered potential biomarkers for the prognosis, diagnosis, and treatment of MDS, further experimental and clinical validation is necessary. In the future, our methodology must undergo additional verification in larger patient cohorts to ensure its reliability and effectiveness.

## IV. Conclusion

In this study, we successfully identified potential molecular pathways associated with myelodysplastic syndromes (MDS) and screened for potential therapeutic targets. These findings not only validate the existing research foundation but also significantly enhance our understanding of the pathological mechanisms underlying this disease. Furthermore, they present novel avenues for the development of innovative therapeutic strategies, holding the potential to improve treatment outcomes and enhance the quality of life for patients with this disease in the future.

## Author contributions

**Conceptualization:** Peng-fei Han, Yan-hui Yu.

**Data curation:** Chang-sheng Liao, Min-xiao Wang.

**Formal analysis:** Chang-sheng Liao.

**Investigation:** Chang-sheng Liao, Min-xiao Wang, Yu-qin Xie, Xue-qin Wei.

**Methodology:** Chang-sheng Liao, Min-xiao Wang.

**Resources:** Min-xiao Wang, Yu-qin Xie, Xue-qin Wei.

**Software:** Chang-sheng Liao, Min-xiao Wang.

**Supervision:** Peng-fei Han, Yan-hui Yu.

**Validation:** Chang-sheng Liao, Min-xiao Wang.

**Visualization:** Chang-sheng Liao, Min-xiao Wang.

**Writing – original draft:** Chang-sheng Liao, Min-xiao Wang.

**Writing – review & editing:** Chang-sheng Liao, Min-xiao Wang.

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
