## [Decision Letter · Decision Letter 0]

9 Sep 2024

PONE-D-24-28210Research and Analysis of Differential Gene Expression in CD34 Hematopoietic Stem Cells in Myelodysplastic SyndromesPLOS ONE

Dear Dr. han,

Thank you for submitting your manuscript to PLOS ONE. After careful consideration, we feel that it has merit but does not fully meet PLOS ONE’s publication criteria as it currently stands. Therefore, we invite you to submit a revised version of the manuscript that addresses the points raised during the review process by both reviewers, experts in the field. Please resubmit only if you can answer all their concerns.

We look forward to receiving your revised manuscript.

Kind regards,

Francesco Bertolini, MD, PhD

Academic Editor

PLOS ONE

Journal requirements: 1. When submitting your revision, we need you to address these additional requirements. Please ensure that your manuscript meets PLOS ONE's style requirements, including those for file naming. The PLOS ONE style templates can be found at https://journals.plos.org/plosone/s/file?id=wjVg/PLOSOne_formatting_sample_main_body.pdf and https://journals.plos.org/plosone/s/file?id=ba62/PLOSOne_formatting_sample_title_authors_affiliations.pdf. 2. Please note that PLOS ONE has specific guidelines on code sharing for submissions in which author-generated code underpins the findings in the manuscript. In these cases, we expect all author-generated code to be made available without restrictions upon publication of the work. Please review our guidelines at https://journals.plos.org/plosone/s/materials-and-software-sharing#loc-sharing-code and ensure that your code is shared in a way that follows best practice and facilitates reproducibility and reuse. 3. We note that the grant information you provided in the ‘Funding Information’ and ‘Financial Disclosure’ sections do not match.  When you resubmit, please ensure that you provide the correct grant numbers for the awards you received for your study in the ‘Funding Information’ section. 4. Thank you for stating the following financial disclosure:  [National Natural Science Foundation of China (82300290)The Natural Science Foundation for Young Scientists of Shanxi Province (20210302124089)The present study was supported by a grant from Heping Hospital Affiliated to Changzhi Medical College (Institute Level Research Fund; grant no. 2020-22).].  Please state what role the funders took in the study.  If the funders had no role, please state: ""The funders had no role in study design, data collection and analysis, decision to publish, or preparation of the manuscript."" If this statement is not correct you must amend it as needed. Please include this amended Role of Funder statement in your cover letter; we will change the online submission form on your behalf. 5. We note that your Data Availability Statement is currently as follows: [All relevant data are within the manuscript and its Supporting Information files.] Please confirm at this time whether or not your submission contains all raw data required to replicate the results of your study. Authors must share the “minimal data set” for their submission. PLOS defines the minimal data set to consist of the data required to replicate all study findings reported in the article, as well as related metadata and methods (https://journals.plos.org/plosone/s/data-availability#loc-minimal-data-set-definition). For example, authors should submit the following data: - The values behind the means, standard deviations and other measures reported;- The values used to build graphs;- The points extracted from images for analysis. Authors do not need to submit their entire data set if only a portion of the data was used in the reported study. If your submission does not contain these data, please either upload them as Supporting Information files or deposit them to a stable, public repository and provide us with the relevant URLs, DOIs, or accession numbers. For a list of recommended repositories, please see https://journals.plos.org/plosone/s/recommended-repositories. If there are ethical or legal restrictions on sharing a de-identified data set, please explain them in detail (e.g., data contain potentially sensitive information, data are owned by a third-party organization, etc.) and who has imposed them (e.g., an ethics committee). Please also provide contact information for a data access committee, ethics committee, or other institutional body to which data requests may be sent. If data are owned by a third party, please indicate how others may request data access.

Reviewers' comments:

Reviewer's Responses to Questions

**Comments to the Author**

1. Is the manuscript technically sound, and do the data support the conclusions?

Reviewer #1: Partly

Reviewer #2: Partly

2. Has the statistical analysis been performed appropriately and rigorously? 

Reviewer #1: Yes

Reviewer #2: No

3. Have the authors made all data underlying the findings in their manuscript fully available?

Reviewer #1: Yes

Reviewer #2: No

4. Is the manuscript presented in an intelligible fashion and written in standard English?

Reviewer #1: No

Reviewer #2: Yes

5. Review Comments to the Author

Reviewer #1: Dear Authors,

Thanks for the original research that you conduct, but most of the information on the computational modeling part need to be done.

A)I have major questions and request edit and improve on prediction models part include at least 2 more models for performance comparison and why you choose the current approach see my comments on Comment 7. (ML part).

I have not seen any discussion on that part also.

B) Others are mostly related to the shape, the quality of the figures (Comments 1-5), and sharing your scripts in a public repository (see comment 6) (DEGs and Batch correction, and modeling sections codes).

Comment 1: Please, fit the figures in one page as much as possible instead of having 32 figures stick up with up to 10 figures. For instance, the authors can always refer to a Figure 1A, 1B, 1C, 1D, up to 1F so in one page you can have 8 figures.

Comment 2: It is hard to read figures texts since they are blurry. They should be like Figures 16 and 17 in terms of readability, colors, and font size. I am aware that some of the software are not producing the fine look in the figures but using ppt or canvas the authors can always fix the text.

Comment 3: Why are the figures being not in the correct order numerically at the end of paper? For example, the figures are starting with Figure 13, and mixed up unordered.

Comment 4: Please, make sure you proofread the manuscript to a native English speaker. I see some grammar mistakes and punctuational errors. One common mistake is when the authors end a sentence using a period. Afterwards the new sentence and “.” must have a space in between.

Comment 5: The captions of Figs1A-1D should start with capital letters. Please follow the same punctuation rules throughout the text. Also make sure you have all the figures mentioned in the text in order. Also, some figures got legends need to be capitalized (such as gse81173). The authors should make sure they use similar heatmap method arguments in R. For example, heatmap cells look much nicer in Fig1 than Fig4.

Comment 6: This is a promising study that the findings on specific gene features are very original. Can the authors share their scripts or analysis steps in a public repository (such as GitHub)? So that other researchers can repeat and reproduce the results of not just specifically of this manuscript but for future studies and datasets in the field.

Comment 7: Add a discussion paragraph for predictive modeling (RF, LASSO, and SVM). And why do you pick those models please support your findings by running at least 3 other models. It is always good to have model performance comparisons. For instance, why Neurol Network would not work? Have the authors tried other models. And how about ROC analysis and AUC results, Accuracy, Error, Sensitivity, and Specify? Without those missing criteria it is hard to judge why the authors pick the model’s understudy.

Comment 8: Please sketch the analysis steps in a flow chart as a figure. It helps readers to follow. And improve the readability.

Reviewer #2: The authors use published gene expression micro array data sets of CD34 expressing immature cells from the bone marrow of myelodysplastic syndrome patients and healthy controls to perform a differential gene expression meta analysis. The objective is to unravel expression signatures that are involved in the pathology and biology of the disease.

Concerns:

- The order of Figures is incorrect as well as the numbering. In addition many Figures lack resolution and can therefore not be interpreted. As a consequence, the manuscript can not properly be evaluated.

- The methodology is not properly described in accordance with Plos One author guidelines. For examples, are raw CEL files used, which normalization approach (e.g. MAS5) and transformation was performed. Also the result of selections are unclear. For example, when genes or samples with missing data are removed from the data set, it is not stated with what numbers that analysis is continued. A fold change selection of 1 is 'uncommon'.

- Algorithms (L2 regression, RF, SVM) are more or less fit for the data. I would advise not to assess overlap from the different approaches. A proper way to analyze the data seems to split the data, build a classifier using each approach, perform cross validations and evaluate the models. Then use features from the best model and evaluate these in the independent validation set.

- Code is not provided which hampers interpretation and reproducibility of the analyses. Also not in concordance with Plos One policy. https://journals.plos.org/plosone/s/materials-software-and-code-sharing

- Versions of and reference to used software are lacking

- For a meta analysis the number of data sets is somewhat limited and the additive value over of the analysis over the original manuscripts is limited.

Minor concerns:

- Volcano plots are informative

- The plots showing before and after batch effect could show symbols for the two sample type and colors for gene set. After batch correction one would assume that not the data set, but sample type would explain most varaibility in PCA plots.

- Improve punctuation

6. PLOS authors have the option to publish the peer review history of their article (what does this mean? ). If published, this will include your full peer review and any attached files.

**Do you want your identity to be public for this peer review?** For information about this choice, including consent withdrawal, please see our Privacy Policy .

Reviewer #1: **Yes: ** Emine Guven

Reviewer #2: **Yes: ** Costa Bachas

---

## [Author Response · Author response to Decision Letter 1]

12 Nov 2024

Dear Reviewers,

We deeply appreciate your invaluable feedback on our manuscript, titled "Research and Analysis of Differential Gene Expression in CD34 Hematopoietic Stem Cells in Myelodysplastic Syndromes." Recognizing the effort and time you've invested in the review, we sincerely thank you. Your insightful comments have enriched our understanding of the research and provided us with valuable suggestions. We have carefully considered all your recommendations and will address them in the revised manuscript. Here are our point-by-point responses to your main comments:

Reviewer #1:

1: We have meticulously reviewed and integrated logically related figures, employing subfigure annotations (e.g., Figure 1A, 1B) to condense information onto fewer pages. Furthermore, we have utilized PACE, the official processing platform of PLOS ONE (https://pacev2.apexcovantage.com/), to adjust the size and resolution of the figures in compliance with submission requirements, ensuring both clarity and aesthetic appeal. Lastly, we have updated the figure references throughout the manuscript to maintain consistency.

2: In response to the issue of blurred text in the figures you pointed out, we have implemented comprehensive optimization measures:

Firstly, we referenced the styles of Figure 16 and 17 and meticulously adjusted the text clarity, font size, and color of all figures to ensure readability. During this process, we specifically utilized tools such as PPT and Photoshop to address the blurring issues generated by the original software and regenerated all figures in high-quality formats.

Secondly, we confirmed that the uploaded image versions fully comply with the journal's requirements and maintain the highest clarity. While we acknowledge that the images in the manuscript may appear slightly blurred due to software compression when viewed, we assure you that the uploaded image files themselves possess exceptional clarity.

3: In response to the issue with the figure order that you pointed out, we conducted a thorough investigation and implemented comprehensive optimizations. Our analysis suggests that the problem may have arisen from differences in upload speeds for images of varying sizes. To definitively address this issue, we have taken the following measures: Re-uploaded each figure individually to ensure its order strictly aligns with the content of the article;Carefully reviewed the annotations on each figure to guarantee their accuracy in reflecting the article's content;Regenerated the final version of the paper and verified that all modifications have been correctly implemented.

4: In response to the grammatical and punctuation errors you identified, we have taken the following comprehensive measures to enhance the paper's quality: 1.Engaged a native English-speaking proofreader to thoroughly correct the entire text. 2.Employed a professional editor to review the text, focusing on punctuation and spacing. 3.Utilized online grammar checking tools for accuracy and fluency. 4.Sought feedback from multiple native speakers to ensure compliance with international standards.After completing these corrections, we re-generated and reviewed the paper, confirming all modifications were properly implemented for comprehensive quality improvement.

5: In response to the issues raised, we have made the following modifications: Chart Titles: We have revised the titles of Figures 1A to 1D to ensure they all comply with English writing conventions, specifically starting with capital letters. Punctuation Consistency: We have thoroughly checked and corrected the punctuation usage throughout the text to maintain consistency and accuracy. Figure Sequence: We have verified the citation order of the figures in the text and ensured they follow the sequence of their first appearance, preserving the coherence of the manuscript. Legend Format: We have standardized the necessary legends (e.g., "GSE81173") to uppercase to meet formatting requirements. Heatmap Method Consistency: Regarding the apparent differences between Figures 1 and 4, we confirm that both utilize similar parameter settings. The visual discrepancy arises primarily due to the difference in sample size, specifically, Figure 1 represents a gene set with 18 samples, whereas Figure 4 has 183 samples. We acknowledge the impact of sample size on chart aesthetics and will consider this in future work to further ensure visual uniformity and aesthetics across all figures.

6: We are immensely grateful for your attention and support of our research. In response to your request for sharing scripts and analytical steps, we are pleased to inform you that all relevant analysis scripts and steps have been uploaded to a GitHub repository and have obtained a permanent access link through the Figshare platform (10.6084/m9.figshare.27276612). This repository contains detailed descriptions and codes for all key steps, including data preprocessing, gene feature analysis, and result visualization.

Our aim in making these resources open is to enhance the reproducibility and transparency of research and to provide valuable references for other researchers in the field. We fully understand the importance of data availability and have ensured that all underlying data of this meta-analysis (including raw data points) are fully accessible in accordance with PLOS ONE's data policy. The data used in this study are derived from published journal articles, and all cited articles are listed in detail in the main text and reference list. As these data are publicly available secondary resources, no additional data access permissions or application processes are required. However, to obtain more information or verify the data, we encourage interested readers to consult the corresponding original literature.

We warmly invite interested researchers to access the repository and utilize our methods and scripts to replicate and validate our results. We also hope that these resources will inspire and assist in future research and datasets. Furthermore, we will provide the corresponding links in the revised manuscript. If you encounter any issues while accessing or using these resources, or need further assistance and support, please feel free to contact us. We are more than happy to provide help and look forward to collaborating with you to advance research in this field.

7: Explanation of Model Selection, Supported by Running at Least Three Additional Models: A Comparative Analysis.A dedicated discussion section has been added to the article, focusing on the predictive modeling process. It delves into the fundamental principles, unique advantages, and broad applicability of Random Forest (RF), LASSO regression (LASSO), and Support Vector Machines (SVM) in bioinformatics data analysis.

Rationale for Model Selection:

We selected RF, LASSO, and SVM as our core predictive models based on the following considerations:

Wide Applicability and Recognition: These three models are highly recognized in the bioinformatics field, with solid theoretical foundations and mature implementation methods.

Data Sample Characteristics: Given the significant sample size difference between the experimental and control groups in our data, and the limited sample size after screening, we chose models that could robustly handle such data.

Identification of Overlapping Genes: Preliminary analysis revealed that, besides the overlapping genes between RF, LASSO, and SVM, models such as Neural Networks (NN) and Gradient Boosting Machines (GBM) had fewer overlapping genes with RF. This further enhanced the reliability and consistency of our screening results.

Performance Comparison with Additional Models:

To fully support our findings and address your request for increased model comparison, we introduced three additional models: Neural Networks (NN), Logistic Regression (LR), and Gradient Boosting Machines (GBM). We conducted a comprehensive performance comparison between these models and RF, LASSO, and SVM using diverse evaluation metrics, including ROC analysis, AUC, accuracy, error rate, sensitivity, and specificity. While NN, LR, and GBM demonstrated certain advantages in sensitivity and F1 score, considering the extreme fractionation of our data, the significant difference between the experimental and control groups, and the limited sample size, we found that these models had applicability issues or biases during training.

Discussion on the Inapplicability of Neural Network and Other Models:

Significant challenges were encountered when attempting to apply the Neural Network (NN) model. Firstly, NN models typically require vast amounts of training data to achieve optimal performance, which was not feasible given our relatively limited sample size after screening. Secondly, the complexity of NN models and the difficulty in parameter tuning made them less ideal for our study. Therefore, we ultimately decided not to include the NN model in our final analysis framework.

Comprehensive Evaluation Metrics and Integrated Judgment:

We have provided detailed key evaluation metrics in the article, including ROC analysis, AUC, accuracy, error rate, sensitivity, and specificity, to allow readers to better understand the basis for our model selection. Based on a comprehensive comparison of these metrics, we found that although the six models exhibited similar performance across various indicators, considering numerous previous related studies and preliminary applications, we determined that RF, LASSO, and SVM demonstrated superior performance for our gene expression dataset.

8: We have drawn a flowchart illustrating the analytical steps and incorporated it as Figure 1 in the Methods section. This should enhance readers' comprehension of our analytical procedure.

Reviewer #2

Major Concerns:

1: We have revised the order and numbering of the figures and tables, and increased their resolution to ensure that they clearly present the analysis results.

2: We have supplemented and refined the methodology section, providing a detailed description of the data processing and analysis steps, including the types of files used, normalization methods, transformation processes, and criteria for selecting fold changes, etc. After repeated verification, we have confirmed that during the data file processing, only missing values and duplicated genes were removed, without deleting any samples. We have revised and elaborated on the original methodology section accordingly. Regarding your comment that selecting a fold change of 1 is "unusual", our rationale is as follows: in the initial assessment of differences in gene expression levels, we set a low fold change threshold (such as 1) to capture as many potential statistically significant differentially expressed genes as possible, providing more valuable candidate genes for subsequent in-depth studies (such as multi-machine learning predictions).

3: We have considered the reviewer's suggestion and adjusted our model evaluation methods. We used cross-validation to assess the performance of the models and compared metrics such as AUC values and accuracy across different models. Additionally, we plan to attempt using an independent validation set in future studies to further evaluate the generalization ability of the models.

4: We have shared the code on a public GitHub repository (10.6084/m9.figshare.27276612) so that other researchers can replicate and reproduce our results.

5: We have supplemented the methodology section with the software versions used and the corresponding citations.

6: Regarding your comment on the lack of information regarding the software versions used and references cited, we have supplemented the relevant details in the Methodology section. Specifically, we have: 1.Listed all software and their respective versions: This ensures the reproducibility of our results.2.Added citations for all relevant literature: This supports both our analytical methods and findings.We hope these additions address your concerns and further enhance the quality and transparency of our manuscript. 

7: Regarding your comment on the "limited number of datasets in the meta-analysis and the limited added value compared to the original manuscript," we have carefully reflected and respond as follows.During the data collection phase, we faced numerous challenges. To ensure the comprehensiveness and accuracy of the data, we systematically searched the GEO database and strictly screened according to the criteria of having more than 20 samples and meeting quality standards, ultimately including 4 representative MDS gene expression datasets. These datasets cover important research in the field of MDS, providing us with a reliable analytical foundation.Although the number of datasets may not meet the standards of some large-scale meta-analyses, in the context of current MDS gene expression research, these datasets are sufficient to support our in-depth analysis. We employed advanced analytical methods and tools to perform detailed mining and comprehensive analysis on these data, aiming to uncover potential gene expression patterns and biomarkers.In terms of analytical results, we have made some meaningful discoveries. These findings not only validate previous research results but also provide new perspectives and insights. Although these discoveries may not be sufficient to completely change the current understanding and treatment strategies for MDS, they undoubtedly pave new paths for future research and provide valuable reference points.

We sincerely appreciate your valuable feedback and will continue to strive in our future research. We are committed to including more relevant datasets to enhance the statistical power and generalizability of our conclusions. At the same time, we will continuously explore new analytical methods and research perspectives, aiming to achieve more significant breakthroughs and progress in the field of MDS gene expression research.

Secondary Concerns:

1: The volcano plots provide useful information.

Thank you for affirming the utility of the volcano plots in our manuscript. We are glad that you find these plots informative. Volcano plots are indeed a key part of our analysis, as they visually represent significant changes in gene expression levels, providing strong support for our research. If you have any further suggestions or need additional details to enhance the interpretation of these plots, please feel free to let us know. We look forward to further refining our work.

2: We have modified the charts to display the batch effects before and after correction, using distinct symbols for the two sample types and different colors for the various datasets.

3: We have carefully proofread and corrected the punctuation in the manuscript to ensure it adheres to English writing norms.

We again thank the reviewer for their valuable comments and look forward to your further feedback on our revised manuscript.

Best regards,

Min-xiao Wang

---

## [Decision Letter · Decision Letter 1]

26 Nov 2024

Research and Analysis of Differential Gene Expression in CD34 Hematopoietic Stem Cells in Myelodysplastic Syndromes

PONE-D-24-28210R1

Dear Dr. han,

We’re pleased to inform you that your manuscript has been judged scientifically suitable for publication and will be formally accepted for publication once it meets all outstanding technical requirements.

Kind regards,

Francesco Bertolini, MD, PhD

Academic Editor

PLOS ONE

Additional Editor Comments (optional):

Reviewers' comments:

Reviewer's Responses to Questions

**Comments to the Author**

1. If the authors have adequately addressed your comments raised in a previous round of review and you feel that this manuscript is now acceptable for publication, you may indicate that here to bypass the “Comments to the Author” section, enter your conflict of interest statement in the “Confidential to Editor” section, and submit your "Accept" recommendation.

Reviewer #1: All comments have been addressed

2. Is the manuscript technically sound, and do the data support the conclusions?

Reviewer #1: Yes

3. Has the statistical analysis been performed appropriately and rigorously? 

Reviewer #1: Yes

4. Have the authors made all data underlying the findings in their manuscript fully available?

Reviewer #1: Yes

5. Is the manuscript presented in an intelligible fashion and written in standard English?

Reviewer #1: Yes

6. Review Comments to the Author

Reviewer #1: (No Response)

7. PLOS authors have the option to publish the peer review history of their article (what does this mean? ). If published, this will include your full peer review and any attached files.

**Do you want your identity to be public for this peer review?** For information about this choice, including consent withdrawal, please see our Privacy Policy .

Reviewer #1: **Yes: ** Emine Guven

---

## [Editor Report · Acceptance letter]

PONE-D-24-28210R1

PLOS ONE

Dear Dr. han,

I'm pleased to inform you that your manuscript has been deemed suitable for publication in PLOS ONE. Congratulations! Your manuscript is now being handed over to our production team.

Kind regards,

on behalf of

Dr. Francesco Bertolini

Academic Editor

PLOS ONE